

# Modelling winter organic aerosol at the European scale with CAMx: evaluation and source apportionment with a VBS parameterization based on novel wood burning smog chamber experiments

**Giancarlo Ciarelli[1], Sebnem Aksoyoglu[1], Imad El Haddad[1], Emily A. Bruns[1], Monica Crippa[2], Laurent Poulain[3], Mikko Äijälä[4], Samara Carbone[5], Evelyn Freney[6], Colin O'Dowd[7], Urs Baltensperger[1] and André S. H. Prévôt[1]**

[1]{Paul Scherrer Institute, Laboratory of Atmospheric Chemistry, 5232 Villigen PSI, Switzerland}

[2]{European Commission, Joint Research Centre (JRC), Directorate for Energy, Transport and Climate, Air and Climate Unit, Via E. Fermi 2749, I-21027 Ispra (VA), Italy}

[3]{Leibniz-Institute for Tropospheric Research (TROPOS), Permoserstr. 15, 04318 Leipzig, Germany}

[4]{University of Helsinki, Department of Physics, Helsinki, Finland}

[5]{Institute of Physics, University of São Paulo, Rua do Matão Travessa R, 187, 05508-090 São Paulo, S.P., Brazil}

[6]{Laboratoire de Météorologie Physique (LaMP), CNRS/Université Blaise Pascal, Clermont-Ferrand, France}

[7]{School of Physics and Centre for Climate & Air Pollution Studies, Ryan Institute, National University of Ireland Galway, University Road, Galway, Ireland}

Correspondence to: S. Aksoyoglu (sebnem.aksoyoglu@psi.ch)

**Abstract**

We evaluated a modified VBS (Volatility Basis Set) scheme to treat biomass burning-like organic aerosol (BBOA) implemented in CAMx (Comprehensive Air Quality Model with extensions). The updated scheme was parameterized with novel wood combustion smog chamber experiments using a hybrid VBS framework that accounts for a mixture of wood



burning organic aerosol precursors and their further functionalization and fragmentation in the
atmosphere. The new scheme was evaluated for one of the winter EMEP intensive campaigns
(February-March 2009) against aerosol mass spectrometer (AMS) measurements performed
at 11 sites in Europe. We found a considerable improvement for the modelled organic aerosol
(OA) mass compared to our previous model application with the mean fractional bias (MFB)
reduced from -61% to -29%.
We performed model-based source apportionment studies and compared results against
positive matrix factorization (PMF) analysis performed on OA AMS data. Both model and
observations suggest that OA was mainly of secondary origin at almost all sites. Modelled
secondary organic aerosol (SOA) contributions to total OA varied from 32 to 88% (with an
average contribution of 62%) and absolute concentrations were generally under-predicted.
Modelled primary hydrocarbon-like organic aerosol (HOA) and primary biomass burning-like
aerosol (BBOA) fractions contributed to a lesser extent (HOA from 3 to 30%, and BBOA
from 1 to 39%) with average contributions of 13 and 25%, respectively. Modelled BBOA
fractions was found to represent 12 to 64% of the total residential heating related OA, with
increasing contributions at stations located in the northern part of the domain.
Source apportionment studies were performed to assess the contribution of residential and
non-residential combustion precursors to the total SOA. Non-residential combustion and
transportation precursors contributed about 30-40% to SOA formation (with increasing
contributions at urban and near industrialized sites) whereas residential combustion (mainly
related to wood burning) contributed to a larger extent, around 60-70%.  Contributions to OA
from residential combustion precursors in different volatility ranges were also assessed: our
results indicate that residential combustion gas-phase precursors in the semi-volatile range
contributed from 6 to 30%, with higher contributions predicted at stations located in the
southern part of the domain. On the other hand, higher volatility residential combustion
precursors contributed from 15 to 38% with no specific gradient among the stations.
The new retrieved parameterization, although leading to a better agreement between model
and observations, still under-predicts the SOA fraction suggesting remaining uncertainties in
the new scheme or that other sources and/or formation mechanisms need to be elucidated.





## 1 Introduction


Organic aerosol (OA) comprises the main fraction of fine particulate matter ($PM_1$) (Jimenez et
al., 2009). Even though the sources of its primary fraction (primary organic aerosol, POA) are
qualitatively known, uncertainties remain in terms of the total emission fluxes annually
released into the troposphere (Kuenen et al., 2014). Moreover, the measured OA load largely
exceeds the emitted POA fractions at most measurement sites around the world. A secondary
fraction (SOA), formed from the condensation of oxidized gases with low-volatility on pre-
existing particles, is found to be the dominant fraction of OA (Crippa et al., 2014; Huang et
al., 2014; Jimenez et al., 2009). Such low-volatility products are produced in the atmosphere
when higher volatility organic gases are oxidized by ozone ($O_3$), hydroxyl (OH) radical and/or
nitrate ($NO_3$) radical. The physical and chemical processes leading to the formation of SOA
are numerous, very uncertain and currently under debate (Hallquist et al., 2009; Tsigaridis et
al., 2014; Fuzzi et al., 2015; Woody et al., 2016).
Available long-term measurements might help in elucidating the composition and origin of
OA in different seasons. Canonaco et al. (2015) presented direct evidence for significant
changes in the SOA fingerprint between summer and winter from 13 months of OA
measurements conducted in Zürich using the aerosol chemical speciation monitor (ACSM).
Their results indicate that summer oxygenated OA mainly arises from biogenic precursors
whereas winter oxygenated OA is more strongly influenced by wood burning emissions.
Moreover, numerous ambient studies of open burning plumes from aircraft do not show a net
increase in OA, despite observing oxidation (Cubison et al., 2011; Jolleys et al., 2012). It is
therefore necessary that the chemical transport models (CTMs) correctly reproduce OA
concentrations by taking into account all the uncertainties and variability of observations.
Most of the CTMs account for common biogenic and anthropogenic high volatility SOA
precursors such as terpenes, isoprene, xylene and toluene which have a saturation
concentration ($C^*$) higher than $10^6$ µg m$^{-3}$ (Aksoyoglu et al., 2011; Ciarelli et al., 2016a). A
few models also include intermediate volatility organic compounds (IVOCs) with a $C^*$ of $10^3$
- $10^6$ µg m$^{-3}$ and semi-volatile organic compounds (SVOCs) with a $C^*$ of 0.1 - $10^3$ µg m$^{-3}$ co-
emitted with POA (Bergström et al., 2012; Ciarelli et al., 2016a; Denier van der Gon et al.,
2015; Fountoukis et al., 2014; Tsimpidi et al., 2010; Woody et al., 2016). In these
applications, the volatility distributions of POA and IVOCs emissions are based on the study



of Robinson et al. (2007), where the IVOC mass is assumed to be 1.5 times the total organic
mass available in the semi-volatile range.
The standard gridded emission inventories do not yet include SVOCs and their emissions are
still highly uncertain as their measurement is strongly affected by the method used (Lipsky
and Robinson, 2006). A recent study by Denier van der Gon et al. (2015) reported a new
residential wood burning emission inventory including SVOCs, where emissions are higher
by a factor of 2-3 on average than those in the EUCAARI inventory (Kulmala et al., 2011).
The new emission inventory was used in two CTMs (EMEP and PMCAMx) and it improved
the model performance for the total OA (Denier van der Gon et al., 2015). Ciarelli et al.
(2016a) showed that allowing for evaporation of primary organic particles as available in
European emission inventories degraded OA performance (further under-predicted OA but
with POA and SOA components in a better agreement) whereas model performance improved
when volatility distributions that implicitly account for missing semi-volatile material
(increasing POA emissions by a factor of 3) were deployed.
Various modelling studies were performed by increasing POA emissions by a factor of 3 to
compensate for the missing gaseous emissions based on partitioning theory predictions
(Ciarelli et al., 2016a; Fountoukis et al., 2014; Shrivastava et al., 2011; Tsimpidi et al., 2010).
Fig. S1 shows the partitioning of ~1 $\mu$g m$^{-3}$ of POA at different temperatures using the latest
available volatility distribution for biomass burning (May et al., 2013). The ratio between the
available gas and particle phase material in the semi-volatile range is predicted to be roughly
3. This implies that, in these applications, the newly emitted organic mass (POA + SVOCs +
IVOCs) is 7.5 times higher than in original emissions (i.e., OM = (3*POA) + (1.5*(3*POA))).
This indirect accounting of missing organic material could be used in the absence of more
detailed gridded emission inventories, keeping in mind that the amount of higher volatility
compounds was specifically derived from studies conducted with diesel engines (Robinson et
al., 2007).
Along with ambient measurement studies, novel wood burning smog chamber studies provide
more insight into wood burning SOA formation and the nature of its precursors. Bruns et al.
(2016) performed several wood-burning aging experiments in a ~7 m$^{3}$ smog chamber. Using
proton-transfer-reaction mass spectrometry (PTR-MS) they characterized SOA precursors at
the beginning of each aging experiment and found that up to 80% of the observed SOA could
be explained with a collection of a few SOA precursors that are usually not accounted in



regional CTMs (e.g. cresol, phenol, naphthalene). Recently, we used those chamber data to
parameterize a hybrid volatility basis set (Ciarelli et al., 2016b). The results provided new
direct information regarding the amount of wood burning SOA precursors which could be
directly used in CTM applications in the absence of more refined wood burning emissions in
gridded inventories. The box-model application reproduced the chamber data with an error of
approximately 25% on the OA mass and 15% on the O:C ratio (Ciarelli et al., 2016b).
In the current study, the updated volatility basis set (VBS) parameterization was implemented
in the comprehensive air quality model with extensions (CAMx) model, and simulations were
performed in Europe for a winter period in February-March 2009. Results are compared with
previous simulations using the original VBS framework (Ciarelli et al., 2016a) and with
source apportionment data at eleven sites with different exposure characteristics, obtained
using PMF applied to AMS measurements (Crippa et al., 2014).

## 134  2   Method

### 135  2.1   Regional modelling with CAMx

The CAMx version 5.41 with VBS scheme (ENVIRON, 2011; Koo et al., 2014) was used in this
study to simulate an EMEP measurement campaign between 25 February and 26 March 2009
in Europe. The modelling method and input data were the same as those used in the
EURODELTA III (ED III) project, described in detail in Ciarelli et al. (2016a). The model
domain covers Europe with a horizontal resolution of 0.25° x 0.25°. Meteorological
parameters were calculated from ECMWF IFS (Integrated Forecast System) data at 0.2°
resolution. There were 33 terrain-following σ-levels from ~20 m above ground level (first
layer) up to about 350 hPa, as in the original IFS data. For the gas phase chemistry, the
Carbon Bond (CB05) mechanism (Yarwood, 2005). The ISORROPIA thermodynamic model
(Nenes et al., 1998) was used for the partitioning of inorganic aerosols (sulfate, nitrate,
ammonium, sodium and chloride). Aqueous sulfate and nitrate formation in cloud water was
calculated using the RADM algorithm (Chang et al., 1987). Formation and evolution of OA is
treated with a hybrid volatility basis set (VBS) that accounts for changes in volatility and O:C
ratio (Koo et al., 2014) with dilution and aging. Particle size distributions were treated with a
two static mode scheme (fine and coarse). The results presented in this study refer to the fine
fraction ($PM_{2.5}$). We parameterized the biomass burning sets based on chamber data as
described in Ciarelli et al. (2016b).





The anthropogenic emission inventory was made available for the ED III community team by
the National Institute for Industrial Environment and Risks (INERIS) at 0.25° x 0.25°
horizontal resolution. More information regarding the anthropogenic emission inventories are
available in Bessagnet et al. (2014, 2016) and Ciarelli et al. (2016a). Hourly emissions of
biogenic VOCs, such as monoterpenes, isoprene, sesquiterpenes, xylene and toluene, were
calculated using the Model of Emissions of Gases and Aerosols from Nature MEGANv2.1
(Guenther et al., 2012) for each grid cell in the model domain.
**2.2   Biomass burning organic aerosol scheme**
The biomass burning organic aerosol scheme was constrained using recently available wood
burning smog chamber data (Bruns et al., 2016) as described in Ciarelli et al. (2016b). The
model deploys three different basis sets (Donahue et al., 2011) to simulate the emissions of
organics from biomass burning and their evolution in the atmosphere. The first set allocates
fresh emissions into five volatility bins ranging with saturation concentrations between $10^{-1}$
and $10^3$ µg m$^{-3}$ following the volatility distribution and enthalpy of vaporization proposed by
May et al. (2013). In order to include gas-phase organics in the semi-volatile range in the
absence of more detailed inventory data, we used the approach proposed by previous studies
(Shrivastava et al., 2011; Tsimpidi et al., 2010). The second set allocates oxidation products
from SVOCs after shifting the volatility by one order of magnitude. The third set allocates
oxidation products from traditional VOCs (xylene, toluene, isoprene, monoterpenes and
sesquiterpenes) and from non-traditional SOA precursors retrieved from chamber data (~4.75
times the amount of organic material in the semi-volatile range, Ciarelli et al., 2016b).
Primary and secondary semi-volatile compounds react with OH in the gas-phase with a rate
constant of $4\times10^{-11}$ cm$^3$ molec$^{-1}$ s$^{-1}$ (Donahue et al., 2013), which decreases their saturation
concentration by one order of magnitude. No heterogeneous oxidation of organic particles or
oligomerization processes is included in the model. The new model parameterization
described in this study is referred to as VBS_BC_NEW throughout the paper to distinguish
from the previous base case called VBS_BC as given in Ciarelli et al. (2016a).
**2.3   Model evaluation**
The model results for the period between 25 February and 26 March 2009 were compared
with OA concentrations measured by AMS at 11 European sites. Modelled BBOA, HOA and
SOA concentrations were compared with multi-linear engine 2 (ME-2) analysis performed on


AMS data (Paatero, 1999) using source finder (SoFi) (Canonaco et al., 2013; Crippa et al.,
2014). Elevated sites such as Montseny and Puy de Dôme were also included in the analysis
and modelled concentrations for these two sites were extracted from higher layers in order to
minimize the artefacts due to topography in a terrain-following coordinate system. This was
not the case in our previous application, where model OA concentrations were extracted from
the surface layer (Ciarelli et al., 2016a). We assumed POA emissions from SNAP2 (emissions
from non-industrial combustion plants in the Selected Nomenclature for Air Pollution) and
SNAP10 (emissions from agriculture, about 6% of POA in SNAP2) to be representative of
biomass burning like emissions. OA emissions from all other SNAP categories, including
emissions from ships, were compared with HOA-resolved PMF factors. Whilst this could be a
reasonable assumption for HOA-like aerosol, it is probably not the case for BBOA-like
aerosol, as gridded emissions for SNAP2 also include other emission sources (i.e., coal
burning which might be important in eastern European countries like Poland). We could not
resolve our emission inventory to that level and the contribution of coal could not be
separated for these European cites (Crippa et al., 2014) in contrast to China (Elser et al., 2016)
using similar statistical methods. Finally, the SOA fraction was compared to the PMF-
resolved oxygenated organic aerosol (OOA) fraction.
Statistics were reported in terms of mean bias (MB), mean error (ME), mean fractional bias
(MFB), mean fractional error (MFE) and coefficient of determination ($R^2$) (see Table S1 for
the definition of statistical parameters).

## 204    3    Results and discussions

### 205    3.1    Analysis of the modelled OA

Figure 1 shows the average modelled OA concentrations and surface temperature for the
period between 25 February and 26 March 2009. Temperatures were below 0°C in the north,
ranged 5-10°C in central Europe and were above 10°C in the southern part of the domain.
Model performance for surface temperature was evaluated within the ED III exercise and
found to be reproduced reasonably well, with a general under-prediction of around 1°C
(Bessagnet et al., 2014).
A clear spatial variability in the modelled OA concentrations is observed (Fig. 1). Predicted
OA concentrations were higher in eastern European countries (especially Romania and
southern Poland) as well as over northern Italy (8-10 µg m$^{-3}$ on average) whereas they were





lower in the northern part of the domain. A similar spatial distribution of OA concentrations
was also reported by Denier van der Gon et al. (2015) using the EMEP model. Relatively high
OA concentrations over the Mediterranean Sea are mainly of secondary origin due to
enhanced photochemical activity (more details are found in Section 3.2). In addition, the
reduced deposition capacity over water leads to higher OA levels.
The scatter plots in Fig. 2 show the modelled (VBS_BC_NEW) versus measured daily
average OA concentrations at 11 sites in Europe together with the results from our previous
model application (VBS_BC, Ciarelli et al., 2016a) for comparison. The modified VBS
scheme (VBS_BC_NEW) predicts higher OA concentrations compared to our previous study
using the original scheme (VBS_BC) (~ 60% more OA on average at all sites). Statistical
parameters improved significantly (Table 1); the mean fractional bias MFB decreased from -
61% in VBS_BC to -29% in VBS_BC_NEW and the model performance criteria were met
(Boylan and Russell, 2006). The coefficient of determination remained almost unchanged for
OA in the VBS_BC_NEW case ($R^2$=0.58) compared to VBS_BC ($R^2$=0.57) indicating that the
original model was able to similarly capture the OA daily variation, but not its magnitude.
The majority of the stations show an $R^2 \geq 0.4$. Lower values were found for the elevated sites
of Montseny and Puy de Dome ($R^2$=0.17 and $R^2$=0.13, respectively) and also at the Helsinki
site ($R^2$=0.06). In spite of the improvements with respect to earlier studies, modelled OA is
still lower than measured (mean bias MB from -0.1 μg m$^{-3}$ up to -3.1 μg m$^{-3}$) at most of the
sites, with only a slight overestimation at a few locations (MB from 0.3 μg m$^{-3}$ up to 0.9 μg m$^{-}$
$^{3}$).
The observed OA gradient among the 11 sites was reproduced very well (Fig. 3). Both
measured and modelled OA concentrations were highest in Barcelona. Other sites with
concentrations greater than 2 μg m$^{-3}$ were Payerne, Helsinki, Vavihill and Montseny.
Barcelona and Helsinki are both classified as urban stations, which justifies the higher OA
loads due to the anthropogenic activities (e.g. traffic, cooking and heating). Anthropogenic
activities in the area of Barcelona could also affect OA concentrations at Montseny which is
about 40 km away. In the case of Payerne and Vavihill, the relatively high OA concentrations
might be due to residential heating, where wood is largely used as a combustion fuel during
cold periods (Denier van der Gon et al., 2015). For Chilbolton, located not far from London,
this might not be the case: the fuel wood usage in the UK is the lowest in Europe (Denier van
der Gon et al., 2015). Ots et al. (2016) suggested the possibility of missing diesel-related



IVOCs emissions, which might be an important source of SOA in those regions. However,
other studies reported substantial contribution from solid fuel combustion to OA (Young et
al., 2015). In this case, it might be that difficulties in reproducing the OA concentration are
mainly related to the relatively complex area of the site (i.e., close to the English Channel).
An evaluation of diurnal variations of HOA and SOA concentrations for this site showed a
consistent under-prediction of both components (Fig. S2).

### 253    3.2    Analysis of the OA components

The predicted POA spatial distribution (Fig. 4) resembles the residential heating emission
pattern of different countries (Bergström et al., 2012). The highest POA concentrations were
predicted in east European countries, France, Portugal and in northern Italy (~3-5 $\mu$g m$^{-3}$)
whereas they were less than 1 $\mu$g m$^{-3}$ in the rest of the model domain. Very low OA
concentrations in Sweden were already shown by previous European studies. Bergström et al.
(2012) reported that Swedish organic carbon (OC) emissions from the residential heating
sector were lower by a factor of 14 compared to Norway, even though Sweden had much
higher wood usage (60% higher) likely due to underestimation of emissions from residential
heating in the emission inventory.
The spatial distribution of SOA concentrations, on the other hand, is more widespread with a
visible north to south gradient (Fig. 4). Higher SOA concentration were predicted close to
primary emission sources (e.g. Poland, Romania, Po Valley and Portugal) but also in most of
the countries below 50° latitude and over the Mediterranean Sea where higher OH
concentration, reduced deposition capacity and high contribution from long-range transport
are expected (average concentrations around 3-4 $\mu$g m$^{-3}$).
Comparison of results from this study (VBS_BC_NEW) with the earlier one (VBS_BC,
Ciarelli et al., 2016a) suggests that the new VBS scheme predicts higher SOA concentrations
by about a factor of 3 (Fig. 5) and improves the model performance when comparing assessed
OOA from measurements with modelled SOA (Table 3).
POA concentrations, on the other hand, are clustered below 1 $\mu$g m$^{-3}$ except in Barcelona,
showing an $R^2$=0.36, (Fig. 5 and Table 2). Although predicted POA concentrations at
Barcelona were lower than the measurements, MFB=-47% and MFE=69% were still in the
range for acceptable performance criteria (MFE ≤+75% and −60 < MFB < + 60 %, Boylan
and Russell, 2006). On the other hand, the model over-predicted the POA concentrations at





Hyytiälä (MFB=131% and MFE=131%), Helsinki (MFB=95% and MFE=100%) and Cabauw
(MFB=76% and MFE=86%) mainly due to the overestimated BBOA fraction as seen in Fig.

280    6.

At most of the sites, OA was dominated by SOA (Fig. 6 and Fig. 7) which was
underestimated in particular at Chilbolton, Melpitz and Vavihill (Table 3). As already
mentioned, the under-prediction of SOA concentrations might be attributed to missing SOA
precursors or uncertainties in SOA formation mechanisms and removal processes. On the
other hand, the remote station of Mace Head showed a positive bias for SOA (MFB = 30%),
even though model and measurement concentrations were very similar (0.54 and 0.35 μg m$^{-3}$,
respectively), which could be attributed to an overestimated contribution from the boundaries.
The relatively small positive bias at the two elevated sites, Montseny and Puy de Dome (MFB
= 4% and 17%, respectively), is most likely the result of difficulties in capturing the inversion
layer.
Mostly traffic-related HOA was underestimated at the urban site Barcelona (Table S2, Fig. 6),
with the model not able to reproduce the diurnal variation of HOA at this urban site likely due
to poorly reproduced meteorological conditions or too much dilution during day time in the
model (Fig. S2). The under-prediction of the HOA fraction is consistent with our previous
study where model evaluation for $NO_2$ revealed a systematic under-estimation of the
modelled concentration (Ciarelli et al., 2016a). The course resolution of the domain (0.25° x
0.25°) may result in too low emissions especially at urban sites. The majority of the $NO_x$
($NO+NO_2$) emissions in Europe arises from the transportation sector (SNAP7), which might
have much larger uncertainties than previously thought (Vaughan et al., 2016). An evaluation
of planetary boundary layer height (PBLH) within the EDIII shows that although the PBLH
was quite well represented in general in the ECMWF IFS meteorological fields, CAMx tends
to underestimate the night-time minima and to overestimate some daytime peaks. The other
urban site considered in this study is Helsinki. In this case, HOA concentrations were over-
predicted, as seen in Figs. 6 and S2, which might indicate missing dispersion processes in the
model or under-estimated dilution.
The modelled BBOA fraction on the other hand was generally higher than the measurements,
with an average MFB of 50%  (Table S3, Figs. 6-7), which might arise from various factors:
1) In the model, POA emissions from SNAP2 and SNAP10 are assumed to be representative
of BBOA emissions which might not be the case for all European countries (other non-wood



fuels such as coal, which is allocated to SNAP2 category and could not be separated in this
study), 2) The under-prediction of the modelled surface temperature (Bessagnet et al., 2014)
will directly influence the partitioning of organic material in the semi-volatile range,
favouring freshly emitted organic material to condense more to the particle phase, 3)
Uncertainties in the adopted volatility distributions and/or in the oxidation processes of semi-
volatile organic vapours.
The temporal variability of OA concentrations was reproduced quite well: most of the peaks
were captured accurately (Fig. 8); the magnitudes of only a few (Vavihill, Chilbolton and
Barcelona) were underestimated. Diurnal variations of HOA, BBOA and SOA components at
the rural-background sites suggest that the model was able to reproduce the relatively flat
profile of the measured SOA and the increased BBOA concentrations at night (Fig. 9). On the
other hand, there was a slight underestimation of HOA during the day, especially around
noon, likely as a result of too much dilution in the model.
In our previous application, we performed a sensitivity study with increased biogenic and
residential heating emissions by a factor of two (Ciarelli et al., 2016a). While the model was
rather insensitive to the increased biogenic emissions during winter periods, a substantial
increase in the OA concentrations was observed when emissions from residential heating
were doubled. The model with doubled emissions from residential heating
(VBC_BC_2xBBOA), overestimated the POA fraction at most of the sites (Fig. 10) with
smaller effects on SOA, even though a better closure was achieved between modelled and
observed OA. The results of the simulations using the new parameterization
(VBC_BC_NEW), on the other hand, were closer to the measurement data especially for the
SOA fraction (Fig. 10).

### 3.3  Residential versus non-residential combustion precursors

More detailed source apportionment studies were performed in order to assess the importance
of residential and non-residential combustion precursors for OA and SOA. The upper panel in
Fig. 11 shows the relative contributions to SOA from residential and non-residential
combustion precursors. The model results indicate that non-residential combustion and
transportation precursors contribute about 30-40% to SOA formation (with increasing
contribution at urban and near-industrialized sites) whereas residential combustion (mainly
related to wood burning) contribute to a larger extent, i.e., around 60-70%. The residential





combustion precursors were further apportioned to semi-volatile and higher volatility
precursors (Fig. 11, lower panel). In particular, SVOC precursors exhibit a south-to-north
gradient with increasing contribution to the residential heating related OA for stations located
in the southern part of the domain (maximum and minimum contributions of 42 and 17% in
Montseny and Hyytiälä, respectively). Such a gradient also reflects the effect of temperature
on the partitioning of semi-volatile organic material: the lower temperatures in the northern
part of the domain will reduce the saturation concentration of the organic compounds
allowing primary organic material to favour the particle phase and reducing the amount of
SVOCs available that could act as SOA precursors. In the southern part of the domain, the
higher temperature will favour more organic material in the semi-volatile range to reside in
the gas-phase, rendering it available for oxidation. On the other hand, no south-to-north
gradient was predicted for the higher volatility class of precursors. Source apportionment for
different volatilities classes of the non-residential and transportation sectors is currently not
implemented for this model application.
A comprehensive summary of the contribution to the total OA from all the sources (i.e. HOA,
BBOA, residential combustion semi-volatile precursors, residential combustion higher
volatility precursors and non-residential combustion precursors) is shown in Fig. 12 at each of
the measurement sites. Residential combustion precursors in the semi-volatile range
contributed from 6 to 30% whereas higher volatility compounds contributed to a larger extent,
i.e. from 15 to 38%. SOA from non-residential combustion precursors contributed from 10 to
37% to the total OA. The primary sources HOA and BBOA contributed from 3 to 30% and 1-
39%, respectively. These results lead to the conclusion that the overall contribution of
residential combustion to OA concentrations in Europe varies between 52% at stations in the
UK and 75-76% at stations in Scandinavia.
**Conclusion**
This study aims to evaluate recent VBS parameterizations in commonly used CTMs and to
underline the importance of taking into account updated and more detailed SOA schemes as
new ambient and chamber measurements elucidate the high complexity and strong variability
of OA. In this context, a new VBS parameterization (based on recent wood burning
experiments) implemented in CAMx was evaluated against high-resolution AMS
measurements at 11 sites in Europe during February-March 2009, one of the winter EMEP
intensive measurement campaigns. Results obtained from this study were compared with





those from our earlier work in which the original VBS scheme in CAMx was applied. A
detailed source apportionment for the organic aerosol (OA) fraction was discussed. This study
provided the following outcome:
-    A considerable improvement was found for the modelled OA concentrations

compared to our previous studies mainly due to the improved secondary organic

aerosol (SOA) performance. The average bias for the 11 AMS sites decreased by

about 60% although the model still underestimates the SOA fraction.

-    Both model and PMF source apportionment based on measurements suggested that

OA was mainly of secondary origin with smaller primary contribution, with primary

contribution of 13 and 25% for HOA and BBOA, respectively. The model

performance for the HOA fraction was reasonably good at most of the sites except at

the urban Barcelona site which could be related to the uncertainties in emissions or too

much dilution in the model. On the other hand, the modelled BBOA was higher than

the measurements at several stations indicating the need for further studies on

residential heating emissions, their volatility distribution and oxidation pathway of the

semi-volatile organic gases.

-    Emissions from the residential heating sector (SNAP2) largely influenced the OA

composition. The modeled primary BBOA fraction contributed from 46% to 77% of

the total primary organic fraction (POA), with an average contribution of 65%. Non-

residential combustion and transportation precursors contributed about 30-40% to

SOA (with increasing contribution at urban and near-industrialized sites) whereas

residential combustion (mainly related to wood burning) contributes to a larger extent,

~ 60-70%. Moreover, the contribution to OA from residential combustion precursors

in different range of volatilities was also investigated: residential combustion gas-

phase precursors in the semi-volatile range contributed from 6 to 30% with a positive

south-to-north gradient. On the other hand, higher volatility residential combustion

precursors contributed from 15 to 38% showing no specific gradient among the

stations.

401



**Acknowledgements**

We thank the EURODELTA III modelling community, especially INERIS, TNO as well as ECMWF for providing various model input data. Calculations of land use data were performed at the Swiss National Supercomputing Centre (CSCS). We thank D. Oderbolz for developing the CAMxRunner framework to ensure reproducibility and data quality among the simulations and sensitivity tests. M. Tinguely for the visualization software, and RAMBOLL ENVIRON for their valuable comments. This study was financially supported by the Swiss Federal Office of Environment (FOEN). The research leading to these results received funding from the European Community's Seventh Framework Programme (FP7/2007-2013) under grant agreement no. 290605 (PSI-FELLOW), from the Competence Center Environment and Sustainability (CCES) (project OPTIWARES) and from the Swiss National Science Foundation (WOOSHI grant 140590). We thank D.A. Day for analysis on the DAURE dataset. Erik Swietlicki for the Vavihill dataset, Claudia Mohr for the Barcelona dataset, A. Kiendler-Scharr for Cabauw AMS data, Eriko Nemitz for the Chilboton data, Karine Sellegri for the Puy de Dôme dataset and Jose-Luis Jimenez for the measurements data in Montseny. The AMS measurements were funded through the European EUCAARI IP. We would like to acknowledge EMEP for the measurement data used here, HEA-PRTLI4, EPA Ireland, and the Science Foundation Ireland for facilitating measurements at Mace Head.



431

432

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





**4 Tables and Figures**

Table 1. Statistics of OA for the VBS_BC_NEW case for February-March 2009 at each AMS
site as well as an average of all sites for both VBS_BC_NEW and VBS_BC. Bold numbers
represent the stations were model performance criteria were met.

| Site* | Mean observed OA ($\mu g\ m^{-3}$) | Mean modelled OA ($\mu g\ m^{-3}$) | MB ($\mu g\ m^{-3}$) | ME ($\mu g\ m^{-3}$) | MFB [-] | MFE [-] | r | $R^2$ |
|---|---|---|---|---|---|---|---|---|
| Barcelona (BCN) | 8.3 | 5.1 | -3.1 | 3.7 | **-0.4** | **0.5** | 0.6 | 0.4 |
| Cabauw (CBW) | 1.2 | 1.5 | 0.3 | 0.7 | **0.1** | **0.5** | 0.7 | 0.4 |
| Chilbolton (CHL) | 2.4 | 1.0 | -1.4 | 1.5 | -0.9 | 0.9 | 0.8 | 0.6 |
| Helsinki (HEL) | 2.7 | 3.6 | 0.9 | 1.8 | **0.3** | **0.6** | 0.3 | 0.1 |
| Hyytiälä (SMR) | 1.3 | 1.7 | 0.3 | 0.8 | **-0.1** | **0.6** | 0.8 | 0.6 |
| Mace Head (MHD) | 0.8 | 0.7 | -0.1 | 0.3 | **-0.1** | **0.7** | 0.7 | 0.5 |
| Melpitz (MPZ) | 1.5 | 0.8 | -0.6 | 0.9 | -0.6 | 0.7 | 0.6 | 0.3 |
| Montseny (MSY) | 3.1 | 3.5 | 0.4 | 2.0 | **0.1** | **0.6** | 0.4 | 0.1 |
| Payerne (PAY) | 4.1 | 2.9 | -1.2 | 1.9 | **-0.5** | **0.7** | 0.7 | 0.4 |
| Puy de Dôme (PDD) | 0.6 | 1.1 | 0.4 | 0.8 | 0.3 | 0.8 | 0.4 | 0.2 |
| Vavihill (VAV) | 3.9 | 2.1 | -1.8 | 2.0 | -0.8 | 0.8 | 0.8 | 0.6 |
| VBS_BC_NEW | 3.0 | 2.3 | -0.7 | 1.6 | **-0.3** | **0.7** | 0.8 | 0.6 |
| VBS_BC (Ciarelli et al., 2016a) | 3.0 | 1.4 | -1.5 | 1.8 | -0.6 | 0.8 | 0.8 | 0.6 |

* Model OA concentrations extracted at surface level except for the stations of Puy de Dôme
and Montseny.






Table 2. Statistics of POA for the VBS_BC_NEW case for February-March 2009 at each
AMS site as well as an average of all sites for both VBS_BC_NEW and VBS_BC. Bold
numbers represent the stations were model performance criteria were met.

| Site | Mean observed POA ($\mu$g m$^{-3}$) | Mean modelled POA ($\mu$g m$^{-3}$) | MB ($\mu$g m$^{-3}$) | ME ($\mu$g m$^{-3}$) | MFB [-] | MFE [-] | r | R$^2$ |
|---|---|---|---|---|---|---|---|---|
| Barcelona | 4.0 | 2.0 | -2.1 | 2.4 | **-0.5** | **0.7** | 0.4 | 0.2 |
| Cabauw | 0.4 | 0.9 | 0.5 | 0.5 | 0.8 | 0.9 | 0.5 | 0.2 |
| Chilbolton | 1.0 | 0.5 | -0.5 | 0.5 | **-0.6** | **0.7** | 0.8 | 0.6 |
| Helsinki | 0.8 | 2.5 | 1.7 | 1.7 | 1.0 | 1.0 | 0.2 | 0.0 |
| Hyytiälä | 0.1 | 0.5 | 0.4 | 0.4 | 1.3 | 1.3 | 0.5 | 0.3 |
| Mace Head | 0.2 | 0.1 | -0.1 | 0.2 | 0.5 | 1.0 | 0.2 | 0.1 |
| Melpitz | 0.3 | 0.3 | 0.1 | 0.2 | **0.3** | **0.7** | 0.5 | 0.2 |
| Montseny | 0.5 | 0.4 | 0.0 | 0.3 | **0.2** | **0.7** | 0.3 | 0.1 |
| Payerne | 0.7 | 1.1 | 0.3 | 0.6 | **0.5** | **0.7** | 0.5 | 0.3 |
| Puy de Dôme | 0.2 | 0.3 | 0.1 | 0.2 | 0.5 | 0.9 | 0.2 | 0.1 |
| Vavihill | 1.1 | 1.0 | -0.1 | 0.6 | **-0.3** | **0.7** | 0.5 | 0.2 |
| VBS_BC_NEW | 0.9 | 0.9 | -0.1 | 0.7 | 0.3 | 0.8 | 0.6 | 0.3 |
| VBS_BC (Ciarelli et al., 2016a) | 0.9 | 0.9 | 0.0 | 0.8 | 0.3 | 0.8 | 0.6 | 0.4 |












Table 3. Statistics of SOA for the VBS_BC_NEW case for February-March 2009 at each
AMS site as well as an average of all sites for both VBS_BC_NEW and VBS_BC. Bold
number represents the stations were model performance criteria were met.

| Site | Mean observed SOA ($\mu$g m$^{-3}$) | Mean modelled SOA ($\mu$g m$^{-3}$) | MB ($\mu$g m$^{-3}$) | ME ($\mu$g m$^{-3}$) | MFB [-] | MFE [-] | r | R$^2$ |
|---|---|---|---|---|---|---|---|---|
| Barcelona | 4.4 | 3.2 | -1.2 | 1.6 | **-0.4** | **0.5** | 0.7 | 0.5 |
| Cabauw | 1.0 | 0.6 | -0.4 | 0.6 | -0.7 | 0.9 | 0.7 | 0.4 |
| Chilbolton | 1.4 | 0.5 | -0.9 | 1.0 | -1.1 | 1.2 | 0.7 | 0.5 |
| Helsinki | 1.8 | 1.1 | -0.7 | 1.1 | -0.7 | 0.9 | 0.4 | 0.2 |
| Hyytiälä | 1.2 | 1.1 | -0.1 | 0.7 | -0.7 | 1.0 | 0.8 | 0.6 |
| Mace Head | 0.4 | 0.5 | 0.2 | 0.6 | 0.3 | 1.0 | 0.4 | 0.2 |
| Melpitz | 1.2 | 0.5 | -0.7 | 0.8 | -1.0 | 1.1 | 0.6 | 0.4 |
| Montseny | 2.6 | 3.1 | 0.5 | 1.8 | **0.0** | **0.7** | 0.4 | 0.1 |
| Payerne | 3.7 | 2.0 | -1.7 | 2.1 | -0.8 | 0.9 | 0.5 | 0.3 |
| Puy de Dôme | 0.6 | 0.9 | 0.3 | 0.8 | 0.2 | 0.9 | 0.2 | 0.1 |
| Vavihill | 2.8 | 1.1 | -1.7 | 1.7 | -1.2 | 1.2 | 0.8 | 0.7 |
| VBS_BC_NEW | 2.1 | 1.4 | -0.6 | 1.2 | -0.6 | 0.9 | 0.7 | 0.5 |
| VBS_BC (Ciarelli et al., 2016a) | 2.1 | 0.5 | -1.5 | 1.6 | -1.1 | 1.3 | 0.7 | 0.6 |










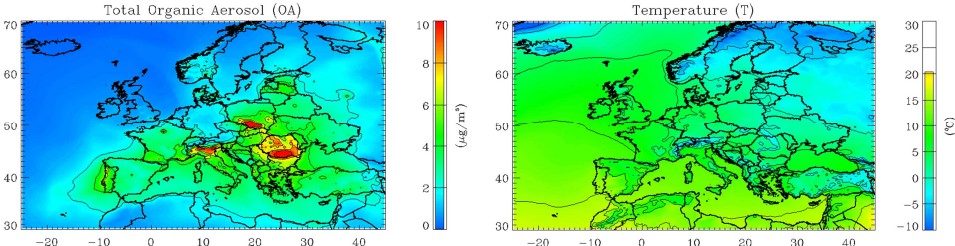


Figure 1. Modelled average total organic aerosol (OA) concentrations (VBC_BC_NEW) and
surface temperature (T) for the period between 25 February and 26 March 2009.


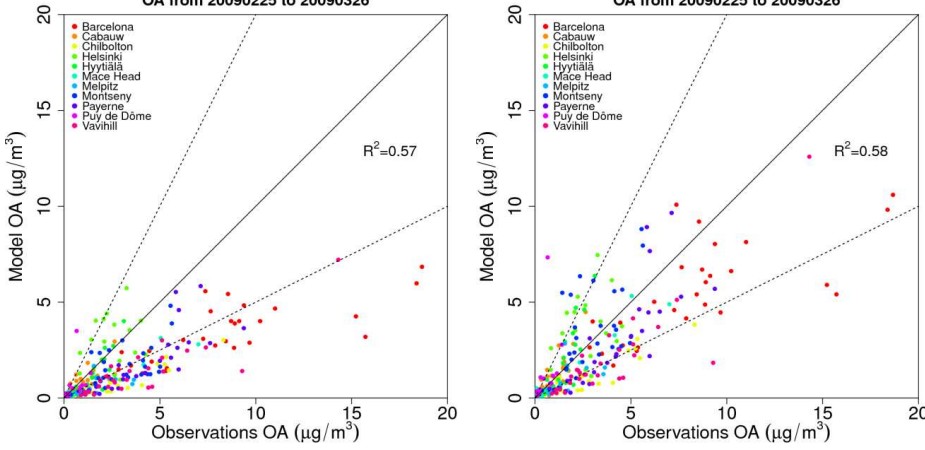


Figure 2. Daily average scatter plots for OA concentrations at 11 AMS sites for the period
between 25 February and 26 March 2009 for VBS_BC (left) and VBS_BC_NEW case (right).
Solid lines indicate the 1:1 line. Dotted lines are the 1:2 and 2:1 lines.



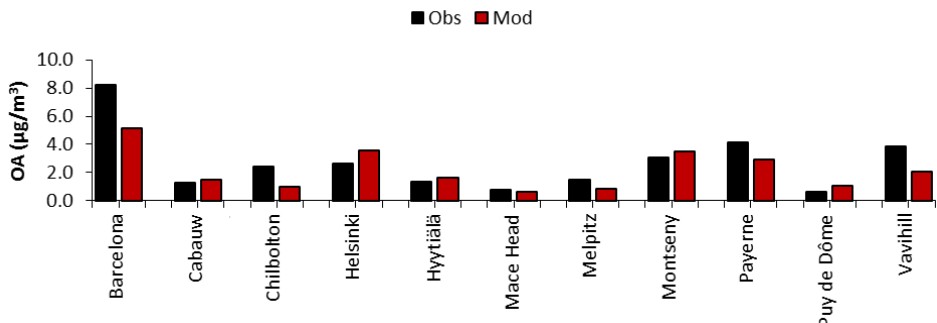


Figure 3. Observed (black) and modelled (VBS_BC_NEW) (red) average OA mass at AMS

sites for the period between 25 February and 26 March 2009.


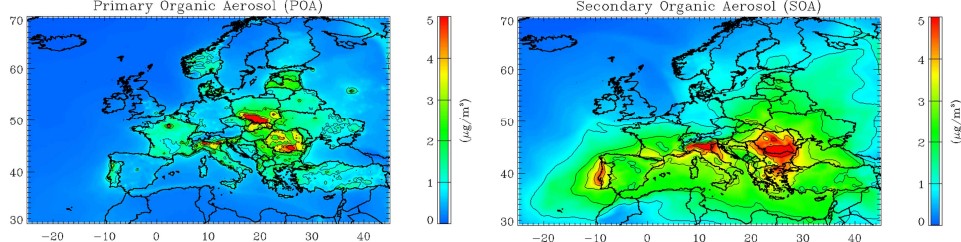


Figure 4. Modelled average POA (left) and SOA (right) concentrations for the period between

25 February and 26 March 2009.








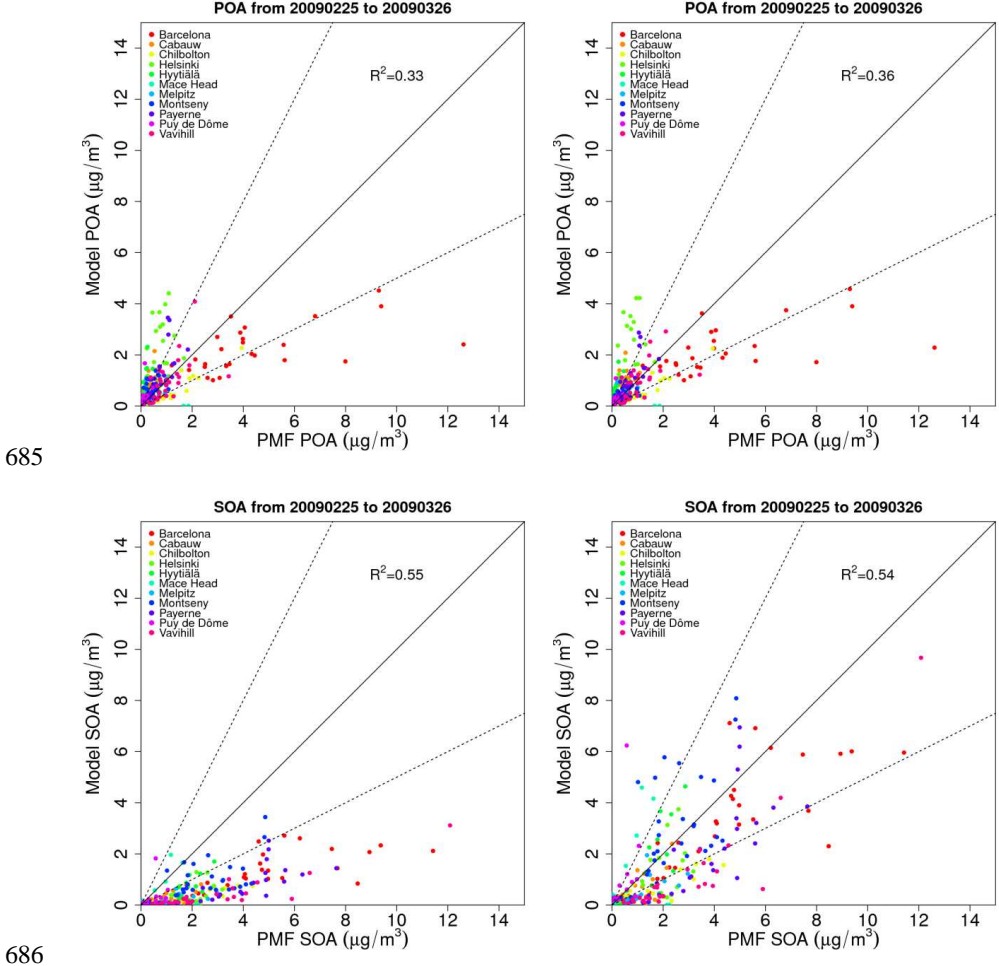



Figure 5. Daily average scatter plots of POA and SOA concentrations at 11 AMS sites for
February-March 2009 in VBS_BC (Ciarelli et al., 2016a) (left) and VBS_BC_NEW (right).
Solid lines indicate the 1:1 line. Dotted lines are the 1:2 and 2:1 lines.





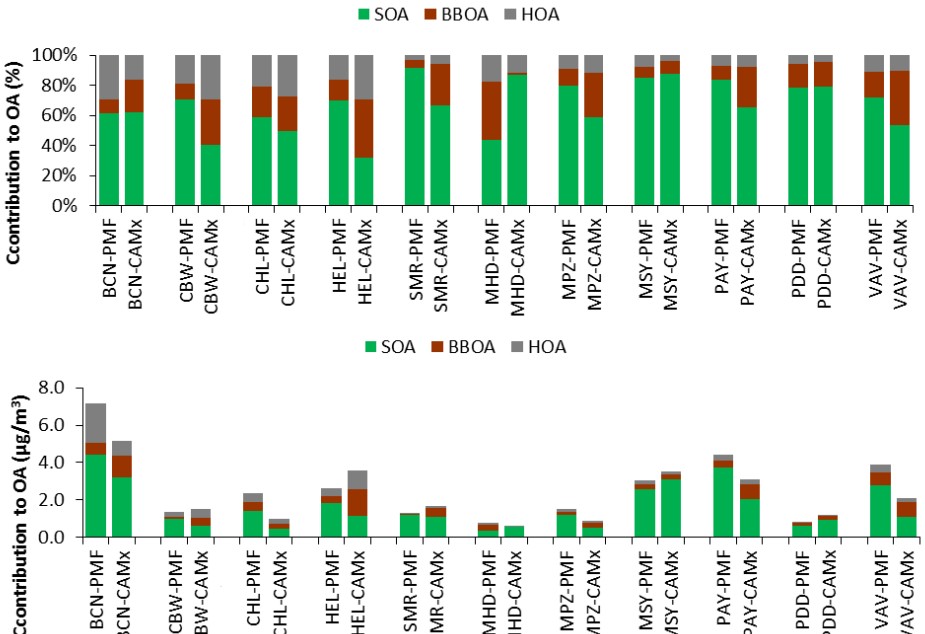



Figure 6. Relative (upper panel) and absolute (lower panel) contribution of HOA, BBOA and SOA to OA concentrations at 11 sites from PMF analysis of AMS measurements (first bar) and CAMx VBS_BC_NEW results (second bar) for the period between 25 February and 26 March 2009.

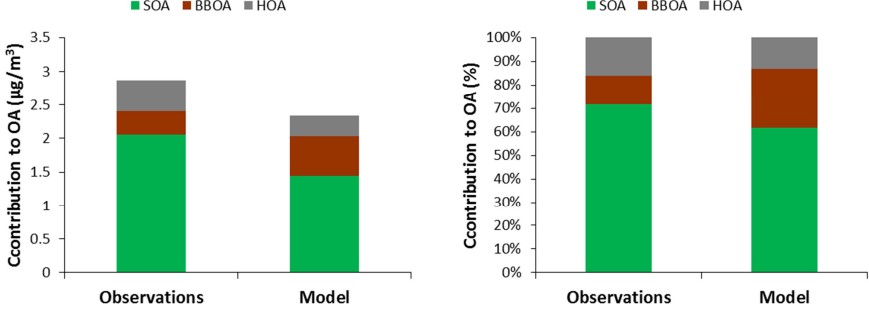


Figure 7. Measured and modelled average absolute (left panel) and relative (right panel) contributions of HOA, BBOA and SOA to OA concentrations for all the 11 sites for the period between 25 February and 26 March 2009.




















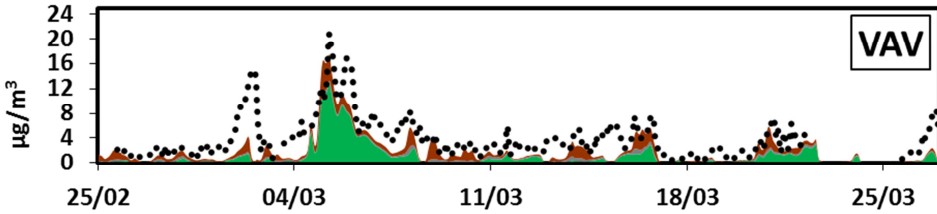


Figure 8. Comparison of measured hourly OA mass concentrations (AMS-OA dotted line),
with modelled components HOA, BBOA and SOA.

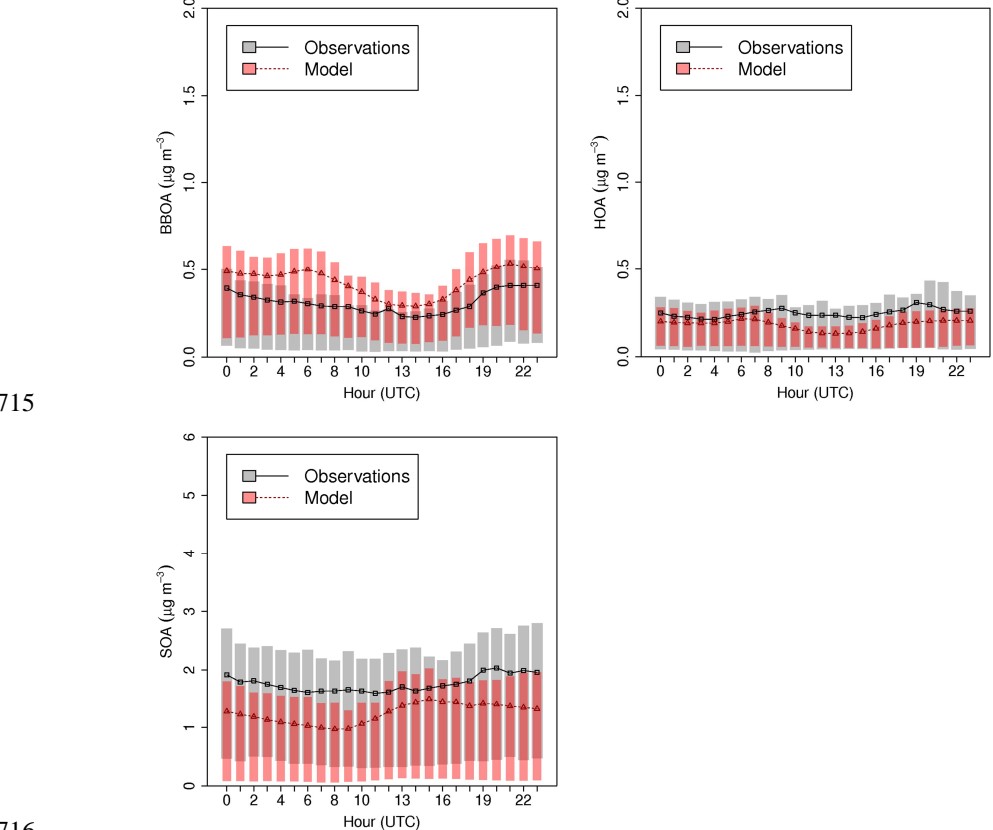



Figure 9. Comparison of modelled (red) and measured (grey) BBOA, HOA and SOA diurnal
profiles at the rural-background sites. The extent of the bars indicates the 25[th] and 75[th]
percentiles.






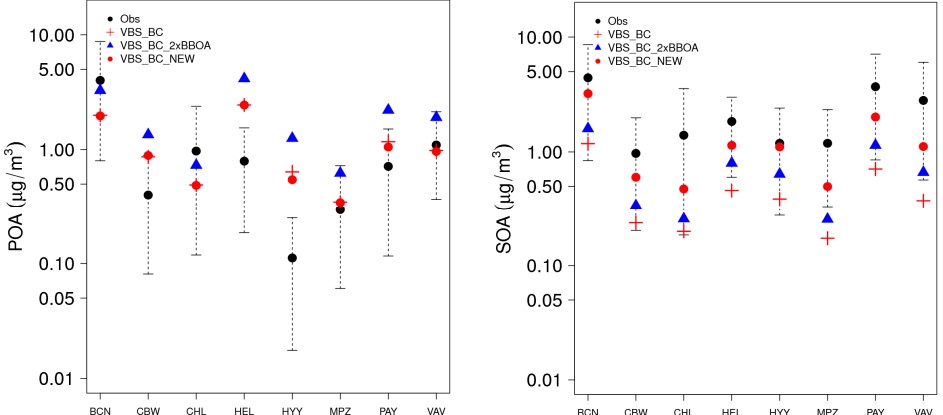



Figure 10. POA (left) and SOA (right) average concentrations at 8 AMS sites for February-
March 2009 in the VBS_BC , VBS_BC_2xBBOA and VBS_BC_NEW cases. Dotted lines
indicate the $10^{th}$ and $90^{th}$ quartile range. Data for the Puy de Dôme and Montseny sites at
higher layers are not available for the VBS_BC_2xBBOA scenario.





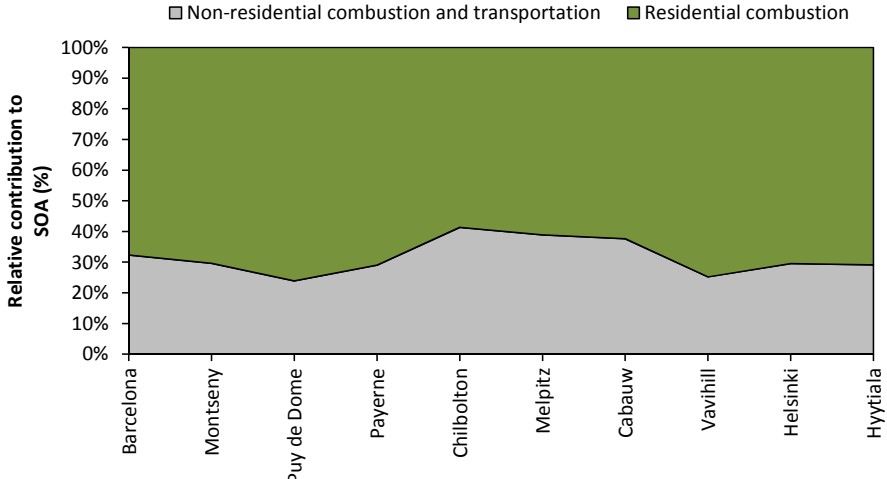


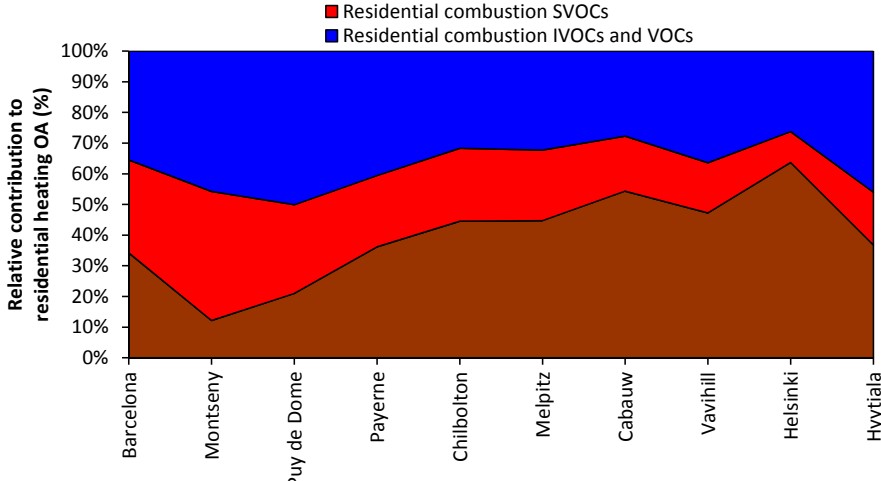


Figure 11. Contribution of residential and non-residential combustion precursors to SOA at
different sites (upper panel). Contribution of BBOA, SVOCs and higher volatility organic
precursors to residential heating OA (lower panel). Stations are ordered from south to north.





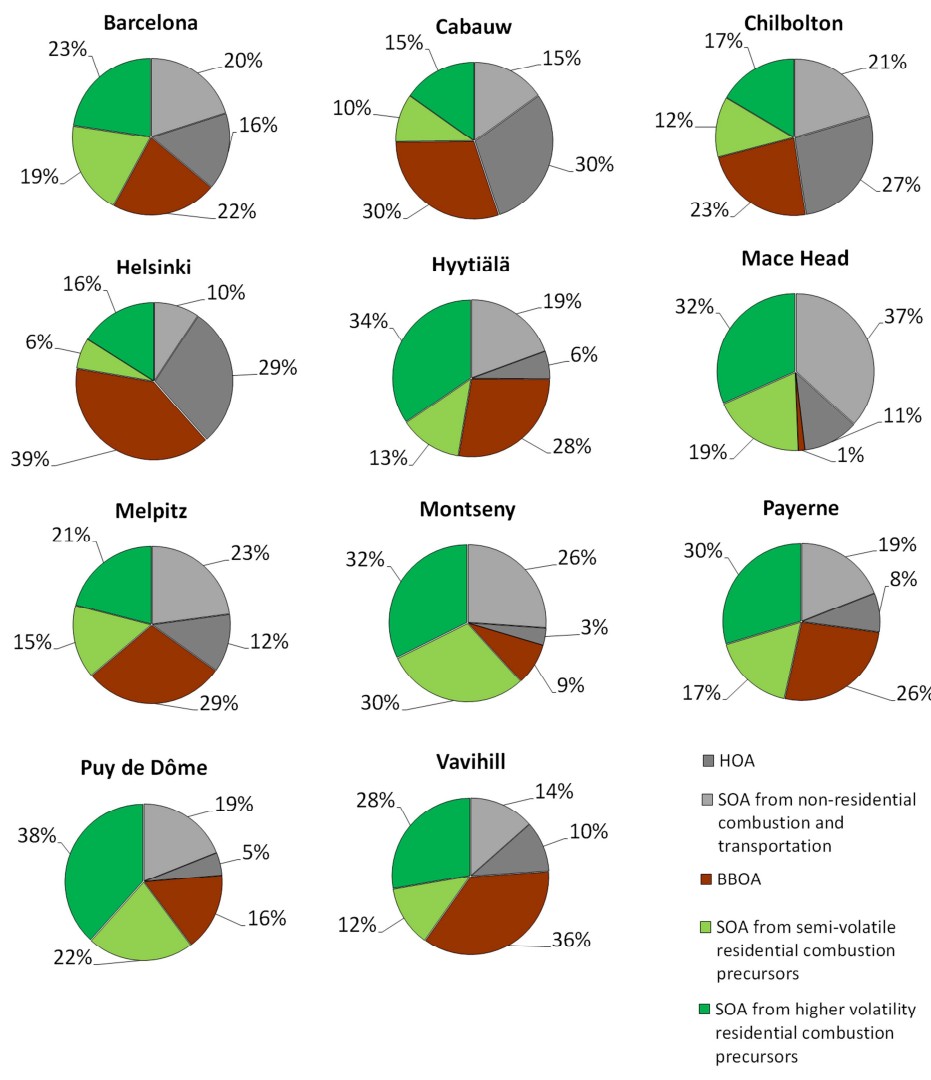


Figure 12. Average modelled composition of OA at the 11 AMS sites for the period between
25 February and 26 March 2009.