# Peer review of "Modelling winter organic aerosol at the European scale"

_Atmospheric Chemistry and Physics, 2016_

## Referee Comment (RC1) · Anonymous Referee #1 · 22 Nov 2016

In the manuscript by Ciarelli et al. a modified VBS scheme for biomass burning-like OA is implemented in the chemistry transport models CAMx. The new VBS scheme was developed by the authors and is described in a paper under review in the GMD (Ciarelli et al., 2016b). CAMx is then used to simulate the wintertime OA mass concentration in Europe in Feb-Mar 2009. The model results are compared with model results from simulations using a different VBS scheme for biomass bring-like OA (Ciarelli et al., 2016a) and with AMS measurements from 11 stations around Europe. The manuscript addresses a very relevant and important topic: the contribution of small-scale residential (mainly wood) combustion to the OA mass loading in Europe during wintertime. My review mainly focuses on the method section, which I partly found quite hard to understand. I have a few critical questions conserving how the different VBS methods was applied which would like to get answered and also explained clearly in the manuscript. If this is done properly and the new VBS parameterization is considered to be scientifically sound by the reviewers of the GMD manuscript Ciarelli et al., 2016b, then I think the manuscript can be suitable for publication in Atmospheric Chemistry and Physics.

**More general comments mainly concerning the method:**
Is it correct that the only difference between the base case model runs from Ciarelli et al., 2016a and this new study is that you use the new VBS sets (called VBS_BC_NEW) instead of VBS_BC to describe the OA formation from biomass burning emissions? If the answer on this question is Yes, which I hope is the case, then please state this clearly in the manuscript. If the answer is No, you have to clearly explain all differences between the two different model runs.

Is it correct that you in total use 3 VBS sets to describe OA formation from biomass burning, 1 set for fresh HOA from fossil fuel combustion, 1 set for aged oxidized HOA, 1 VBS set for BVOC oxidation products (no ageing considered). Thus in total 6 VBS sets? I think you need to describe this more clearly in the manuscript and how this compares to the VBS sets used in Ciarelli et al., 2016a.

In Ciarelli et al. (2016a) for further aging of ASOA and POA vapors from HOA-like emissions you use a reaction rate with OH of $2 \times 10^{-11}$ $cm^3$ $molec^{-1}$ $s^{-1}$. How about this new study? Was it $4 \times 10^{-11}$ $cm^3$ $molec^{-1}$ $s^{-1}$ instead or is this only the reaction rate used for BBOA precursors?

I suggest that you create a table where you list all VBS sets used in the two different model simulations and if they represent SOA or POA, the sources (i.e. BVOCs, biomass burning or fossil fuel burning) and if they represent HOA, BBOA or Biogenic OA. As the manuscript is written now I get very confused about which organic compounds that are POA and SOA, their origin and if they are classified as HOA, BBOA or Biogenic OA. If I understand it correctly the BBOA is only the not atmospheric processed (oxidized) POA emission from biomass burning, and the POA from biomass burning sources that evaporates and then form more oxidized OA is treated as SOA (VBS_BC_NEW set 2) and the VBS_BC_NEW set 3 OA is always treated as SOA. I think you need to more clearly state that BBOA is only referring to the POA from biomass burning but not the SOA formed from biomass burning. I suggest that you change from BBOA to BBPOA. It is only in the abstract L40-41 that you mention that BBOA is referring to primary biomass burning-like OA and HOA primary hydrocarbon-like OA. I missed this and was confused about this when I was

reading the manuscript the first time.

The new version of the model underestimates the OA to a less extent than the previous version. The only difference between the model runs is how the BBOA formation is treated. On L306-307 you write: *"The modelled BBOA fraction on the other hand was generally higher than the measurements, with an average MFB of 50% (Table S3, Figs. 6-7)"*.
I interpret this as that the model improves the modeled total OA but at least partly for the wrong reason because it gives too much BBOA.

In Ciarelli et al. (2016a) where you use the VBS_BC method you write: *"Further aging of BSOA is not considered in this study, based on previous modelling results showing overprediction of OA when such process is taken into account (Lane et al., 2008; Murphy and Pandis, 2009). This implies that also further aging of POA vapors from BBOA-like emissions was not considered since it is performed in the same basis set."*
This is a large assumption which I think might be one of the main reason why you get much less OA (especially SOA from biomass burning sources) when you run VBS_BC instead of VBS_BC_NEW. This needs to be discussed and explained in the manuscript. I also think that you should run a model simulation using VBS_BC but separating the POA vapors from BBOA and allow them to be further oxidized in the same way as the HOA vapors. Then you compare the model results from this method with the model simulations with your new VBS sets (VBS_BC_NEW). To me this is a more fair evaluation of your new biomass burning-like OA VBS parameterizations compare to the old parameterization from Koo et al. (2014) that to my understanding allowed oxidation of evaporated POA from biomass burning sources. If your new biomass burning VBS parameterization still gives substantially better agreement with the observations than the Koo et al. (2014) VBS parameterization, then your contribution to the field can be considered substantial and important.

On L49-54 you write: *"Contributions to OA from residential combustion precursors in different volatility ranges were also assessed: our results indicate that residential combustion gas-phase precursors in the semi-volatile range contributed from 6 to 30%, with higher contributions predicted at stations located in the southern part of the domain. On the other hand, higher volatility residential combustion precursors contributed from 15 to 38% with no specific gradient among the stations."*
I don't understand how you can separate the SVOC molecules in the gas-phase from the SVOC molecules in the particle phase. If we assume equilibrium partitioning between a condensed liquid phase and the air, then the same molecule species are cycled back and forth between the gas- and particle phase because of evaporation and condensation. Do you mean: additional OA formed because of SVOC oxidation in the gas-phase as compared to additional OA formed as a consequence of IVOC oxidation in the gas-phase?
Related to this I also wonder if all POA species (which are SVOCs) are allowed to evaporate (assuming equilibrate with the gas-phase at all times) and form more oxidized organic compounds that become SOA in the model when they re-condense. Thus, is it correct that all POA species eventually end up as more oxidized SOA species in the model? Or is only a fraction of the POA species allocated to the gas-phase and the rest is fixed in the particles based on the initial fresh POA VBS distribution (Fig S1)?

On L156-159 you write: "*Hourly emissions of biogenic VOCs, such as monoterpenes, isoprene, sesquiterpenes, xylene and toluene, were calculated using the Model of Emissions of Gases and Aerosols from Nature MEGANv2.1 (Guenther et al., 2012) for each grid cell in the model domain.*"
But into what VBS scheme are the BVOC oxidation products added? In Koo et al., (2014) and Ciarelli et al. (2016a) you have a 4[th] VBS set for BVOC oxidation products which are not allowed to age because then you get too much SOA.
According to Section 3.3 all of the modeled SOA can either be attributed to residential or non-residential combustion. What about the SOA from BVOCs? Did you not consider BVOCs when you calculated the SOA formation in this new study? I thought that the only difference between the base case model run in Ciarelli et al. (2016a) and in this work was the VBS setup for the BBOA and it's transformation to SOA? This needs to be clarified. At least for the southernmost stations I would expect that BSOA formation also is substantial during the wintertime, and transport from south to north could bring this SOA to the northern latitudes too.

On L170-173 you write: "*The third set allocates oxidation products from traditional VOCs (xylene, toluene, isoprene, monoterpenes and sesquiterpenes) and from non-traditional SOA precursors retrieved from chamber data (~4.75 times the amount of organic material in the semi-volatile range, Ciarelli et al., 2016b).*"
So do I understand it correctly that this 3[rd] VBS set for VOCs originating from biomass burning only considers the traditional VOCs emitted from biomass burning but not the traditional VOCs from other sources? I.e. the same traditional VOC species but from other sources (vegetation and fossil fuel) is added to other separate VBS sets. This, would be desirable since it allows you to distinguish SOA formed from biomass burning, biogenic VOCs and VOCs from fossil fuel sources?

On L232-235 you write: "In spite of the improvements with respect to earlier studies, modelled OA is still lower than measured (mean bias MB from -0.1 μg m$^{-3}$ up to -3.1 μg m$^{-3}$) at most of the sites, with only a slight overestimation at a few locations (MB from 0.3 μg m-3 up to 0.9 μg m-3)."
Here I think you also should mention that the model might underestimate the OA formation because no gradual BVOC oxidation is considered. Or maybe even more if you did not consider any biogenic SOA formation?

Was the influence of NO considered when you divided the SOA precursors into the different VBS bins as was done by Koo et al. (2014)?

To summarize: The model results looks reasonable and the agreement between the model and observations are as good as you could expect both when using the new VBS set and the old VBS set from Ciarelli et al. (2016a). But to me it still remain to be shown that the new VBS parameterization for biomass burning-like OA substantially improves the model performance as to compared to the VBS parameterization developed by Koo et al. (2014). I.e. you need to compare the model results from the simulations with your new VBS parameterization with a simulation using the Koo et al., (2014) VBS parameterization where you also allow the evaporated BBOA material to be further oxidized in the gas-phase. I also think you need to evaluate if not at least part of the reason why the model underestimates the OA is because it underestimates or maybe not even considers biogenic SOA

formation.

**Minor specific comments:**

L47, Page 1: Here you use the term "*transportation precursors*". I think you mean precursors from the road transportation sector. I think you should change the formulation a bit to make this clearer.

L78-79 You write: "*Moreover, numerous ambient studies of open burning plumes from aircraft do not show a net increase in OA, despite observing oxidation (Cubison et al., 2011; Jolleys et al., 2012).*"
I suggest that you reformulate this sentence and instead write something like:
*Moreover, numerous ambient studies with aircraft of open biomass burning plumes do not show a net increase in OA, despite observed oxidation (Cubison et al., 2011; Jolleys et al., 2012).*
When I first read this sentence I thought the open burning plumes came from the aircraft but then I realized that the aircrafts where only used for the measurements of the open biomass burning plumes.

L98-103: The sentence: "*Ciarelli et al. (2016a) showed that allowing for evaporation of primary organic particles as available in European emission inventories degraded OA performance (further under-predicted OA but with POA and SOA components in a better agreement) whereas model performance improved when volatility distributions that implicitly account for missing semi-volatile material (increasing POA emissions by a factor of 3) were deployed.*" is hard to understand. I suggest that you split it into two or three sentences. What do you mean with "*degraded OA performance*"? Do you mean: degraded the model performance concerning the modeled total OA mass?

On L112-115 you write: "*This indirect accounting of missing organic material could be used in the absence of more detailed gridded emission inventories, keeping in mind that the amount of higher volatility compounds was specifically derived from studies conducted with diesel engines (Robinson et al., 2007).*"
In fact I think the Robinson et al., (2007) study was only performed on one single diesel engine (a single-cylinder Yanmar diesel generator), which I expect do not represent modern diesel car engines very well.
I suggest that you instead of "*diesel engines*" at least write: *a singe diesel engine*.

On L284-287 you write: "*On the other hand, the remote station of Mace Head showed a positive bias for SOA (MFB = 30%), even though model and measurement concentrations were very similar (0.54 and 0.35 μg m-3, respectively), which could be attributed to an overestimated contribution from the boundaries.*"
What do you mean by " *overestimated contribution from the boundaries*"? Is it the influence from the model boundary conditions?

On L337-340 you write: "*The model results indicate that non-residential combustion and transportation precursors contribute about 30-40% to SOA formation (with increasing contribution at urban and near-industrialized sites) whereas residential combustion (mainly related to wood burning) contribute to a larger extent, i.e., around 60-70%.*"
I suggest that you change to:

*The model results indicate that non-residential combustion and transportation precursors contribute to about 30-40 % of the SOA formation (with increasing contribution at urban and near-industrialized sites) whereas residential combustion (mainly related to wood burning) contribute to a larger extent, i.e., around 60-70%.*

On line L349-351 you write: "*In the southern part of the domain, the higher temperature will favour more organic material in the semi-volatile range to reside in the gas-phase, rendering it available for oxidation.*"
I would also expect that the higher UV-light intensity in the south caused more SOA formation because of higher OH concentrations.

On line L351-351 you write: "*On the other hand, no south-to-north gradient was predicted for the higher volatility class of precursors.*"
Do you mean?
*On the other hand, no south-to-north gradient was predicted for the SOA formed from the higher volatility class of precursors.*

On L 291-294 you write: "*Mostly traffic-related HOA was underestimated at the urban site Barcelona (Table S2, Fig. 6), with the model not able to reproduce the diurnal variation of HOA at this urban site likely due to poorly reproduced meteorological conditions or too much dilution during day time in the model (Fig. S2).*"
Can it not also be because of too weak diurnal variations in the HOA emissions from traffic in the model?
Reflection: But in the case of Helsinki it seem as if the model instead gives substantially more HOA during the morning (6 UTC, 8 am local time, and 15 UTC, 5 pm local time), which is what you would expect if the HOA mainly came from the local traffic. But surprisingly to me the observations do not indicate any increased local HOA contribution during the morning and afternoon rush hours in Helsinki. Could it be related to the vehicle fleet in Helsinki (i.e. is the road traffic very much dominated by gasoline cars which do not emit much primary HOA but precursors for SOA formation)?

---

## Referee Comment (RC2) · Anonymous Referee #2 · 16 Dec 2016

General Comments:

Ciarelli et al. follow up two other recent publications by augmenting the CAMx VBS implementation with their new parameterization for emission and aging of BBOA emissions. The study itself is a useful application and soundly conceived. The authors find better model-measurement agreement than their previous implementation, but I am troubled by some aspects of their methods and analysis, as described below. Their inclusion of the factor of 3 multiplier to account for missing SVOCs was an approach

originally recommended for Mexico City but has not been used for Europe by previous EUCAARI model studies (e.g. Fountoukis et al., 2014). I am open to the authors' interpretation/justification for this choice (especially if I've misinterpreted the situation), but on its face this is a rather critical assumption that could put major aspects of the paper's conclusions in jeopardy. Moreover, the application of modeled PM2.5 mass to PM1.0 measurements raises questions about how much of the model agreement is spurious. Considering both of these potential biases together, it is concerning that the model predictions for SOA and POA are still lower in many cases than the VBS predictions published by Fountoukis et al. (2014) for the same model scenario. I could recommend this paper for publication after these issues are resolved.

Specific Comments:

1. Page 4, line 108-113: The ratio of semivolatile to nonvolatile material is, as the authors know, a function of the emission source, fuel, and operating conditions – I think it is overly simplistic and actually unhelpful to state that the ratio is predicted to be "roughly 3." The Shrivastava et al. (2011) and Tsimpidi et al. (2010) studies argued that those SVOCs at Mexico City were missing from the inventories because the emissions were parameterized using ambient observations of OA, which would have already equilibrated to atmospheric conditions. On the other hand, the emission factors used to inform the gridded inventories of Europe and the US are, to my knowledge, derived from laboratory scale tests, where much of those SVOCs are notoriously condensed in the particle phase in undiluted exhaust. My reading of Fountoukis et al. (2014) does not lead me to believe that they enhanced their SVOC emissions by a factor of 3 over POA. Rather, I believe they simply repartitioned the existing POA, and they added an additional 1.5*POA for the IVOCs as the authors state. Ciarelli et al. (2016a) shows that the extra SVOCs are needed to improve the model performance (i.e. VBS_BC did much better than VBS_ROB), but I disagree that there is evidence that SVOCs are underestimated in European inventories by so much. Instead, I would argue the real source of this mass is still unknown and is probably a combination of underestimated

[Figure]

SOA yields, aqueous processing, aging of anthropogenic and biogenic SOA and some missing SVOCs as well.

At minimum, a considerable amount of rewriting in the methods, conclusions and abstract is necessary so that the authors communicate explicitly that an unknown fraction of these SVOCs are very likely double-counted and that this parameter needs to be refined and probably lowered in the future as more explicit pathways are added to the model.

2. I agree with the first reviewer that there needs to be significant more description of the VBS framework used here. The diagrams in Ciarelli et al. (2016b) are helpful and there should be a table or diagram in this manuscript that summarize that information for the entire VBS picture including emissions and aging.

3. What is being done about wildfires in the model? Were there any during the EU-CAARI scenario? Are they represented well in the emissions inputs? If so, how do they effect the source apportionment analysis that is presented?

4. On page 5, lines 150-151, the authors point out that CAMx is predicting PM2.5. But the evaluation is against AMS observations which I presume are primarily PM1.0. Doesn't this fact make the frequent underprediction in SOA even more troubling? Is anything more specific known about the diameter of PM2.5 particles to allow the authors to estimate the fraction that would be PM1.0 and thus more applicable to the measurements?

5. Given that points 1 and 4 would lead one to expect substantial overprediction by the model, please also explain why the current predictions are lower than those in Fountoukis et al. (2014) at many sites.

6. Page 9, lines 269-272: This discussion of Fig. 5 is very light. If there is not more to discuss, I recommend removing the figure and just stating the improvement in MB and r.

[Figure]

7. How does the BBOA doubling sensitivity case fit in the context of the VBS_BC_NEW case which is multiplied by 3 and then by 1.5 again? What fraction of that total added vapor mass makes it into the particle phase? This is related to point 8.

8. The description and discussion of BBOA aging should be expanded. Please summarize the aging process as described in Ciarelli et al. (2016b). How is this similar/different to the aging of the traditional biogenic SOA? I assume the authors are not using the Koo et al. (2014) approach where the BBOA ages once and then stops? What is the fractional contribution of the various volatility bins to the total in time and space? Do they actually need 4 VBS bins to represent the aging, or would just using one bin and an IVOC precursor also work reasonably well? Why did they not use the O:C obtained from these AMS data to constrain the aging of the BBOA or the SOA?

Minor Issues/Typos

1. Page 2, line 53: What do the authors mean by "higher volatility?" Are these IVOCs or VOCs? And do they mean that the products of these and the semivolatile precursors contributed 15 to 38%?

2. Page 3, line 62: Consider replacing "qualitatively" with "nominally." They are very similar for sure but while qualitatively to me suggests one knows a lot about the relative importance of each source (just not the actual numbers), nominal suggests you just know that the sources are there and you can name them. The latter to me is more representative of our knowledge of sources for SOA.

3. Page 3, lines 65-71: Please also mention aqueous-phase formation and the importance of solubility in water somewhere here to make the picture more complete.

4. Page 3, line 82: Consider removing the word "common." And refer to SOA explicitly here. For example: "Most CTMs today account for SOA formation from biogenic and anthropogenic. . . A few models also include SOA formation from intermediate volatility„".".

5. I don't think you need a hyphen in "semi-volatile" anywhere in the text, but this is your preference.

6. Page 4, line 114-115: The higher volatility emission parameters were also constrained using monitoring network measurements in the previous modeling studies. Several studies have played with 1.5 factor for instance and it has remained as the parameter of choice despite uncertainties.

7. Page 7, lines 193-199: I was confused by this group of sentences. Consider rewriting for clarity. Maybe something like, "We assumed OA emissions from SNAP2 (emissions from non-industrial combustion plants in the Selected Nomenclature for Air Pollution) and SNAP10 (emissions from agriculture, about 6% of POA in SNAP2), to be representative of biomass burning emissions and thus comparable to the BBOA PMF factor. OA from all other SNAP categories were compared against HOA-like PMF factors. Unfortunately, gridded emissions for SNAP2 include other emission sources (i.e., coal burning which might be important in eastern European countries like Poland). We could not resolve our emission inventory with sufficient detail to separate the contribution of coal for these European cites (Crippa et al., 2014)."

8. Page 8, line 219: Please do not call it deposition "capacity" as this suggests something about the ability of the sea to hold pollution. Please reword. "Efficiency" might make more sense. Or just say "reduced deposition". Also change on page 9, line 267.

9. Page 8, line 236: Please provide some statistic for this statement.

10. Fig. 3: Consider adding error bars to this plot showing variability to make this figure more useful.

11. Page 9, lines 258-262: This sentence needs to be split into two sentences and reworded for clarity.

12. Page 10, line 288-290: Do you have evidence from other PM species or pollutants to back up this claim?

[Figure]

13. Page 10, line 291-305: This sentence should be revised for clarity. The authors have blamed the meteorology and the host model configuration itself but why not the emissions? The activity data for the emissions could be wrong, or the emission factors could be wrong, no? Ok, CAMx has issues like any other CTM, but what makes the authors so sure that most of the problem is not in the emissions data?

14. Page 10, line 296: course should be spelled coarse

15. Page 10, line 308-315: The authors can also add here the potential double-counting of SVOC emissions and the application of PM2.5 prediction to a (nominal) PM1.0 measurement.

16. Page 11, line 316-318: How many of the peaks were captured well? What statistic determines how well they were captured? Unless this statement can be quantified, please remove it.

17. Page 11, line 322: Please consider changing "likely" to "possibly."

18. Figure 10. Please consider using median values in these plots rather than averages. 1) It will more effectively reduce the influence of extreme pollution days. 2) It will be more consistent with your use of percentiles. Consider also adding percentiles for the model run data.

19. Figure 11: This data would be better represented as a bar plot since the x-axis is not really a continuum, even though you are trying to approximate one by ordering them south-north.

20. Tables: please add one more significant figure to all data. I can't figure out why the mean biases are different than the differences in the mean model and mean obs. Is it a rounding issue?

21. Page 13, line 380-388: Please quantify "reasonably good." Compared to what?

22. Figure 11: Is BBOA actually just primary BBOA? Please make this clear in this

figure and throughout the text as it gets confusing.

---

## Author Comment (AC1) · 4 Apr 2017

**Responses to the comments of anonymous referee #1**

Thank you for your comments which helped to improve our manuscript. Please find below your comments in blue, our responses in black and modifications in the revised manuscript in *italic.*

In the manuscript by Ciarelli et al. a modified VBS scheme for biomass burning-like OA is implemented in the chemistry transport models CAMx. The new VBS scheme was developed by the authors and is described in a paper under review in the GMD (Ciarelli et al., 2016b). CAMx is then used to simulate the wintertime OA mass concentration in Europe in Feb-Mar 2009. The model results are compared with model results from simulations using a different VBS scheme for biomass bring-like OA (Ciarelli et al., 2016a) and with AMS measurements from 11 stations around Europe. The manuscript addresses a very relevant and important topic: the contribution of small-scale residential (mainly wood) combustion to the OA mass loading in Europe during wintertime. My review mainly focuses on the method section, which I partly found quite hard to understand. I have a few critical questions conserving how the different VBS methods was applied which would like to get answered and also explained clearly in the manuscript. If this is done properly and the new VBS parameterization is considered to be scientifically sound by the reviewers of the GMD manuscript Ciarelli et al.,2016b, then I think the manuscript can be suitable for publication in Atmospheric Chemistry and Physics.

More general comments mainly concerning the method:

1.
Is it correct that the only difference between the base case model runs from Ciarelli et al., 2016a and this new study is that you use the new VBS sets (called VBS_BC_NEW) instead of VBS_BC to describe the OA formation from biomass burning emissions? If the answer on this question is Yes, which I hope is the case, then please state this clearly in the manuscript. If e answer is No, you have to clearly explain all differences between the two different model runs.

The answer to this question is yes; all model input data prepared for Ciarelli et al. (2016a) were kept the same for this new application (VBS_BC_NEW). The model scheme to treat biomass burning like organic aerosol was updated based on Ciarelli et al. (2016b) which was accepted for final publication in GMD.

2.
Is it correct that you in total use 3 VBS sets to describe OA formation from biomass burning, 1 set for fresh HOA from fossil fuel combustion, 1 set for aged oxidized HOA, 1 VBS set for BVOC oxidation products (no ageing considered). Thus in total 6 VBS sets? I think you need to describe this more clearly in the manuscript and how this compares to the VBS sets used in Ciarelli et al., 2016a.

We agree with the referee that further description of the VBS sets is needed in the manuscript. The model deploys 3 sets to treat biomass burning-like aerosol (as shown in Ciarelli et al., 2016b) and 2 sets to treat HOA-like aerosol, based on Koo et al., (2014). In addition, it assumes that the primary semivolatile vapours from the HOA generate SOA, and not POA, upon oxidation with the OH radical and further condensation in the particle-phase. However, we don't have a separate set to allocate oxidation products from biogenic precursors, and they follow the same oxidation pathways of biomass burning-like aerosol as in the previous case (Ciarelli et al., 2016a), including aging. We are currently working on an updated version of CAMx which includes the separation of biogenic sources. In our reply to comment 8, we present a sensitivity test with no biogenic SOA formation in order to better address the importance of this source.
We further clarified this point as follows:

at line 179 of the revised manuscript

*The third set allocates oxidation products from the traditional VOCs and biogenic precursors (xylene, toluene, isoprene, monoterpenes and sesquiterpenes)*

and at line 185-186:
*This implies that also aging of biogenic products is implicitly taken into account.*
Moreover, we added Table 1, as suggested in comment 4, to clarify all the different sets used in the model (as also suggested by referee #2).

3.
In Ciarelli et al. (2016a) for further aging of ASOA and POA vapors from HOA-like emissions you use a reaction rate with OH of 2 x $10^{-11}$cm³molec$^{-1}$s$^{-1}$. How about this new study? Was it 4 x $10^{-11}$cm³molec$^{-1}$s$^{-1}$ instead or is this only the reaction rate used for BBOA precursors?

A reaction rate of 4 x $10^{-11}$cm³molec$^{-1}$s$^{-1}$ was used to treat aging of biomass semivolatile SOA which we also applied to the rest of anthropogenic sources (referred to as HOA in the manuscript) in order to be consistent among all the other anthropogenic sources and as already proposed by more recent studies for the range of saturation concentrations used here (Donahue et al., 2013; Jo et al., 2013; Hodzic et al., 2016).

We added the following information at line 187 of the revised manuscript:
*A reaction rate of 4 x $10^{-11}$cm³molec$^{-1}$s$^{-1}$ was also applied to the rest of the anthropogenic sources (referred to as HOA ) in order to be consistent among all the other anthropogenic sources as already proposed by more recent studies for the range of saturation concentrations used here (Donahue et al., 2013).*

4.
I suggest that you create a table where you list all VBS sets used in the two different model simulations and if they represent SOA or POA, the sources (i.e. BVOCs, biomass burning or fossil fuel burning) and if they represent HOA, BBOA or Biogenic OA. As the manuscript is written now I get very confused about which organic compounds that are POA and SOA, their origin and if they are classified as HOA, BBOA or Biogenic OA. If I understand it correctly the BBOA is only the not atmospheric processed (oxidized) POA emission from biomass burning, and the POA from biomass burning sources that evaporates and then form more oxidized OA is treated as SOA (VBS_BC_NEW set 2) and the VBS_BC_NEW set 3 OA is always treated as SOA. I think you need to more clearly state that BBOA is only referring to the POA from biomass burning but not the SOA formed from biomass burning. I suggest that you change from BBOA to BBPOA. It is only in the abstract L40-41 that you mention that BBOA is referring to primary biomass burning-like OA and HOA primary hydrocarbon-like OA. I missed this and was confused about this when I was reading the manuscript the first time.

We agree with the referee and added Table1 listing all the sets/sources that we used. We also changed BBOA to BBPOA throughout the manuscript in order to clarify that BBOA refers only to the primary fraction.

*Table 1. Properties of the VBS space. Oxygen numbers for each volatility bin were calculated using the group-contribution of Donahue et al. (2011). Hydrogen numbers were calculated from the van Krevelen relation (Heald et al., 2010).*

|  | log (C*) | Oxygen number | Carbon number | Hydrogen number | Molecular weight |
|---|---|---|---|---|---|
| POA set1* (BBOA-like) Primary biomass burning (BBPOA) | -1 | 4.11 | 11.00 | 17.89 | 216 |
|  | 0 | 3.43 | 11.75 | 20.07 | 216 |
|  | 1 | 2.73 | 12.50 | 22.27 | 216 |
|  | 2 | 2.01 | 13.25 | 24.49 | 216 |

| | | | | | |
|---|---|---|---|---|---|
| | 3 | 1.27 | 14.00 | 26.73 | 215 |
| SOA set2* | -1 | 4.53 | 9.00 | 13.47 | 194 |
| (BBOA-like) | 0 | 4.00 | 9.25 | 14.50 | 189 |
| SOA from SVOCs | 1 | 3.40 | 9.50 | 15.60 | 184 |
| biomass burning | 2 | 2.83 | 9.75 | 16.67 | 179 |
| SOA set3* | -1 | 5.25 | 5.00 | 4.75 | 149 |
| (BBOA-like) | 0 | 4.70 | 5.25 | 5.80 | 144 |
| SOA from | 1 | 4.20 | 5.50 | 6.80 | 140 |
| VOC/IVOCs biomass | 2 | 3.65 | 5.75 | 7.85 | 135 |
| burning and biogenics | 3 | 3.15 | 6.00 | 8.85 | 131 |
| | -1 | 2.69 | 17.00 | 31.3 | 278 |
| POA set1** | 0 | 2.02 | 17.50 | 33.0 | 275 |
| (HOA-like) | 1 | 1.34 | 18.00 | 34.7 | 272 |
| Rest of primary | 2 | 0.63 | 18.5 | 36.4 | 268 |
| anthropogenic sources | 3 | 0.0 | 19.00 | 38.0 | 266 |
| SOA set1** | -1 | 4.90 | 7.00 | 9.10 | 172 |
| (HOA-like) | 0 | 4.38 | 7.25 | 10.1 | 167 |
| SOA from rest of all | 1 | 3.84 | 7.50 | 11.2 | 163 |
| anthropogenic in all | 2 | 3.30 | 7.75 | 12.2 | 158 |
| volatility range | 3 | 2.74 | 8.00 | 13.3 | 153 |
| (SVOCs,IVOCs,VOCs) | | | | | |

*Based on Ciarelli et al. (2016b).
**Molecular structure as in Koo et al. (2014) and Ciarelli et al. (2016a).

5.
The new version of the model underestimates the OA to a less extent than the previous version. The only difference between the model runs is how the BBOA formation is treated. On L306-307 you write: *"The modelled BBOA fraction on the other hand was generally higher than the measurements, with an average MFB of 50% (Table S3, Figs. 6-7)"*. I interpret this as that the model improves the modeled total OA but at least partly for the wrong reason because it gives too much BBOA.

The model improves mainly because more SOA is predicted for the investigated period, whereas statistics for the POA fractions remained almost unchanged (Table 3 and Table 4 in the revised manuscript). The BBPOA fraction remained almost unchanged respect to the VBS_BC scenario (Table S4).
We reformulated the sentence at line 321 of the revised manuscript as below and added Table S4:

*The modelled BBPOA fraction on the other hand was generally overpredicted as in our previous application (Table S4), with an average MFB of 50% (Table S3, Figs. 6-7)*

*Table S4. Comparison of statistics for BBPOA in VBS_BC_NEW with VBS_BC (average of all sites in February-March 2009)*

| | Mean obs ($\mu$g m$^{-3}$) | Mean mod ($\mu$g m$^{-3}$) | MB ($\mu$g m$^{-3}$) | ME ($\mu$g m$^{-3}$) | MFB [-] | MFE [-] |
|---|---|---|---|---|---|---|
| VBS_BC | 0.36 | 0.60 | 0.24 | 0.45 | 0.47 | 0.98 |
| VBS_BC_NEW | 0.36 | 0.59 | 0.23 | 0.43 | 0.50 | 0.97 |

Moreover, PMF analysis is also affected by uncertainties, especially regarding the separation between the BBOA (primary) and SOA (secondary) fractions (Crippa et al., 2013).

6.

*In Ciarelli et al. (2016a) where you use the VBS_BC method you write: "Further aging of BSOA is not considered in this study, based on previous modelling results showing overprediction of OA when such process is taken into account (Lane et al., 2008; Murphy and Pandis, 2009). This implies that also further aging of POA vapors from BBOA-like emissions was not considered since it is performed in the same basis set." This is a large assumption which I think might be one of the main reason why you get much less OA (especially SOA from biomass burning sources) when you run VBS_BC instead of VBS_BC_NEW. This needs to be discussed and explained in the manuscript. I also think that you should run a model simulation using VBS_BC but separating the POA vapors from BBOA and allow them to be further oxidized in the same way as the HOA vapors. Then you compare the model results from this method with the model simulations with your new VBS sets (VBS_BC_NEW). To me this is a more fair evaluation of your new biomass burning-like OA VBS parameterizations compare to the old parameterization from Koo et al. (2014) that to my understanding allowed oxidation of evaporated POA from biomass burning sources. If your new biomass burning VBS parameterization still gives substantially better agreement with the observations than the Koo et al. (2014) VBS parameterization, then your contribution to the field can be considered substantial and important.*

We thank the referee for this comment. We included Figure S4 in the manuscript where we compared the modelled OOA fraction as predicted by VBS_BC, VBS_BC with BBOA vapours allowed to be further oxidized as in Koo et al. (2014) and VBS_BC_NEW. The Koo et al. 2014 VBS approach with BBOA vapours allowed to get further oxidized (Figure S4 middle panel) also helped bringing model and observation in a better agreement, but to a lesser extent compared to VBS_BC_NEW (Figure S4 right panel). In order to emphasize the importance of aging processes we added the following statement at line 244 of the revised manuscript:

*Improvements in the modelled SOA fraction were also observed using the original VBS approach (Koo et al. 2014) when aging of the biomass burning vapours were taken into account (Figure S4).*

[Figure]

*Figure S4. Modelled versus PMF SOA; with VBS_BC (Ciarelli et al., 2016a) (left panel), with VBS_BC where BBOA vapours were allowed to be further oxidized (Koo et al. 2014) (middle panel), and with VBS_BC_NEW (right panel).*

7.

*On L49-54 you write: "Contributions to OA from residential combustion precursors in different volatility ranges were also assessed: our results indicate that residential combustion gas-phase precursors in the semi-volatile range contributed from 6 to 30%, with higher contributions predicted at stations located in the southern part of the domain. On the other hand, higher volatility residential combustion precursors contributed from 15 to 38% with no specific gradient among the stations."*
I don't understand how you can separate the SVOC molecules in the gas-phase from the SVOC molecules in the particle phase. If we assume equilibrium partitioning between a condensed liquid

phase and the air, then the same molecule species are cycled back and forth between the gas-and particle phase because of evaporation and condensation. Do you mean: additional OA formed because of SVOC oxidation in the gas-phase as compared to additional OA formed as a consequence of IVOC oxidation in the gas-phase? Related to this I also wonder if all POA species (which are SVOCs) are allowed to evaporate (assuming equilibrate with the gas-phase at all times) and form more oxidized organic compounds that become SOA in the model when they re-condense. Thus, is it correct that all POA species eventually end up as more oxidized SOA species in the model? Or is only a fraction of the POA species allocated to the gas-phase and the rest is fixed in the particles based on the initial fresh POA VBS distribution (Fig S1)?

Yes, in the sentence we refer to the amount of OA formed due to SVOC oxidation in the gas-phase, and further condensation, and amount of OA formed as a consequence of IVOC oxidation in the gas-phase followed by further condensation.

Not all the POA (SVOCs) species are allowed to evaporate in the model and end up to SOA. The POA species (SVOCs) at $\log_{10}C^*=-1$ is used as a proxy for all non-volatile species and will only reside in the particle phase. We added this information in the caption of Fig. S1 as:

*The lowest bin ($\log_{10}C^*=-1$) is used as a proxy for all non-volatile species which will only reside in the particle phase.*

For the other bins, the amount of SVOCs allocated to the gas-phase depends on the absorptive mass: e.g. a compound with a $C^*=10$ µg m$^{-3}$ will reside 10%, 50% and 90% in the gas phase at $C_{OA} = 100$ µg m$^{-3}$, 10 µg m$^{-3}$ and 1 µg m$^{-3}$, respectively. Likewise, the proportion of this compound in the gas-phase increases with increasing temperature. As a consequence, at lower OA concentrations or at higher temperature, the oxidation of this compound is expected to proceed more rapidly.

We modified the sentence slightly (line 49) in the revised text as:

*Contributions to OA from residential combustion precursors in different volatility ranges were also assessed: our results indicate that residential combustion gas-phase precursors in the semivolatile range (SVOC) contributed from 6 to 30%, with higher contributions predicted at stations located in the southern part of the domain. On the other hand, the oxidation products of higher volatility precursors (the sum of IVOCs and VOCs) contribute from 15 to 38% with no specific gradient among the stations.*

8.
On L156-159 you write: "*Hourly emissions of biogenic VOCs, such as monoterpenes, isoprene, sesquiterpenes, xylene and toluene, were calculated using the Model of Emissions of Gases and Aerosols from Nature MEGANv2.1 (Guenther et al., 2012) for each grid cell in the model domain.*"
But into what VBS scheme are the BVOC oxidation products added? In Koo et al., (2014) and Ciarelli et al. (2016a) you have a 4th VBS set for BVOC oxidation products which are not allowed to age because then you get too much SOA. According to Section 3.3 all of the modeled SOA can either be attributed to residential or non-residential combustion. What about the SOA from BVOCs? Did you not consider BVOCs when you calculated the SOA formation in this new study? I thought that the only difference between the base case model run in Ciarelli et al. (2016a) and in this work was the VBS setup for the BBOA and it's transformation to SOA? This needs to be clarified. At least for the southernmost stations I would expect that BSOA formation also is substantial during the wintertime, and transport from south to north could bring this SOA to the northern latitudes too.

Certainly the SOA formation from BVOCs is very important and it was always considered in all the versions. There is, however, no separate set to allocate oxidation products from biogenic precursors, therefore they follow the same oxidation pathways of biomass burning like aerosol, including aging. We are currently working on an updated version of CAMx in which biogenic sources will be separated. In order to address the question of the referee, we performed a sensitivity test with no biogenic formation (where the reactions of isoprene, monoterpene and sesquiterpene against the OH, O3 and NO3 oxidants were turned off) and compared the results with the base case (VBS_BC_NEW). The contribution of biogenic SOA is then retrieved by calculating the difference between SOA in the

reference simulation (including biogenic SOA formation) and the one with no biogenic SOA formation. Based on this test, BVOC contribution to SOA was predicted to be around 20% for the stations at the lowest latitude (Spain) and a decreasing trend with increasing latitude (less than 5% in Helsinki and Hyytiälä) was found (Figure S5). This is consistent with higher temperatures and consequently more biogenic emissions at locations in the south than those in the north. However, the most predominant source was still predicted to be anthropogenic. We also included the snow cover for March 2009 as retrieved from the TERRA/MODIS instrument in Figure S6. Larger parts of the Scandinavian countries were almost completely covered with snow, partially suppressing the emission of biogenic precursors and in line with less contribution predicted from biogenic sources in Helsinki and Hyytiälä by the model (for the investigated periods).

Comparisons between VBS_BC, VBS_BC_NEW and the sensitivity test with no biogenic SOA formation, showed similar improvement, with differences occurring mainly in the southern stations of Barcelona and Montseny (Figure S7). We revised the legends of Figures 11 and 12 to make it more clear that the biomass burning set also includes biogenic SOA and we added results from the sensitivity test at line 372 of the revised manuscript and at the last bullet point of the conclusions as below:

Line 372:

*Since biogenic SOA is included in the same set as the biomass burning (set3) for this model application, we performed a sensitivity test with no SOA formation from biogenic precursors (where the reactions of isoprene, monoterpene and sesquiterpene with OH, O3 and NO3 were turned off). Our results indicated that for this period, biogenic precursors contribute to SOA to a lesser extent (5-20%) than the anthropogenic ones, with higher contributions at southern stations consistent with higher temperatures, and consequently more biogenic emissions compared to the northern stations (Figure S5). The most predominant source was still predicted to be anthropogenic. Snow cover for March 2009 as retrieved from the TERRA/MODIS revealed that larger parts of the Scandinavian countries were almost completely covered with snow (Fig. S6), partially suppressing the emission of biogenic precursors and in line with very low contribution predicted from biogenic sources in Helsinki and Hyytiälä. Comparison of SOA from VBS_BC_NEW and the sensitivity test with no biogenic SOA formation showed similar improvement with respect to VBS_BC, with differences occurring mainly in the southern stations of Barcelona and Montseny (Fig. S7).*

In the conclusions as last bullet point:

- *Model simulation performed with and without biogenic SOA formation revealed that, for this period, biogenic SOA contributed only to a small extent to the total SOA (5-20%), with an increasing gradient from north to south.*

[Figure]

*Figure S5. Biogenic and anthropogenic contribution to SOA at stations from south to north retrieved as a difference between the predicted SOA in the reference simulation (including biogenic) and a sensitivity test with no biogenic SOA formation.*

[Figure]

.

*Figure S6. Snow cover for March 2009 as retrieved by the TERRA/MODIS instrument.*

[Figure]

*Figure S7. Modelled versus PMF SOA; with VBS_BC (Ciarelli et al., 2016a) (left panel), with VBS_BC_NEW (middle panel), and with VBS_BC_NEW but without biogenic SOA (right panel).*

9.
On L170-173 you write: "*The third set allocates oxidation products from traditional VOCs (xylene, toluene, isoprene, monoterpenes and sesquiterpenes) and from non-traditional SOA precursors retrieved from chamber data (~4.75 times the amount of organic material in the semi-volatile range, Ciarelli et al., 2016b).*"So do I understand it correctly that this 3rd VBS set for VOCs originating from biomass burning only considers the traditional VOCs emitted from biomass burning but not the traditional VOCs from other sources? I.e. the same traditional VOC species but from other sources (vegetation and fossil fuel) is added to other separate VBS sets. This, would be desirable since it allows you to distinguish SOA formed from biomass burning, biogenic VOCs and VOCs from fossil fuel sources?

For this application we don't have a separate set to allocate oxidation products from biogenic precursors, and they follow the same oxidation pathways of biomass burning like aerosol, including aging. Fossil and non-fossil emissions are separated but biomass combustion from residential heating and biogenic precursors are treated in the same set (as in Koo et al., 2014). We strongly agree with the referee that a separate set, exclusively for biogenic precursors, would be highly desirable and we are currently working on an updated version of CAMx which includes such separation. The sensitivity test mentioned above suggests that BVOC emissions might contribute to the total SOA mass by ~5-20%.

10.
On L232-235 you write: "*In spite of the improvements with respect to earlier studies, modelled OA is still lower than measured (mean bias MB from -0.1 µg m-3up to -3.1 µg m-3) at most of the sites, with only a slight overestimation at a few locations (MB from 0.3 µg m-3 up to 0.9 µg m-3).*"Here I think you also should mention that the model might underestimate the OA formation because no gradual BVOC

oxidation is considered. Or maybe even more if you did not consider any biogenic SOA formation?

As we stated earlier, we do consider SOA formation from biogenic precursors and they follow the same oxidation pathways of biomass burning like aerosol, including aging and as we wrote above, we strongly agree with the referee that a separate set, exclusively for biogenic precursors, would be highly desirable. However there could also be other reasons for the under-prediction of the modelled SOA fraction presented here (as also addressed in the reply to referee #2). Marine OA emissions are not included in our simulation. Gantt et al., (2015) showed that primary marine organic aerosol has a weaker coastal-to-inland concentrations gradient than sea-salt aerosol with some inland European cities having more than 10% of the submicron organic aerosol mass as a marine source. Fire emissions, which were not included for this study, are likely to be less important for this comparison since there were few fires activity data during the considered periods (as also addressed in the reply to referee 2). Moreover, aqueous phase SOA formation is not considered in this model application, which might be important for explaining the remaining discrepancies between model and retrieved OOA from measurements.

11.
Was the influence of NO considered when you divided the SOA precursors into the different VBS bins as was done by Koo et al. (2014)?

Yes, the influence of NO was considered as in Koo et al. (2014) and it was based on smog chamber data (Murphy and Pandis, 2009 and Hildebrandt et al., 2009).

To summarize: The model results looks reasonable and the agreement between the model and observations are as good as you could expect both when using the new VBS set and the old VBS set from Ciarelli et al. (2016a). But to me it still remain to be shown that the new VBS parameterization for biomass burning-like OA substantially improves the model performance as to compared to the VBS parameterization developed by Koo et al. (2014). I.e. you need to compare the model results from the simulations with your new VBS parameterization with a simulation using the Koo et al., (2014) VBS parameterization where you also allow the evaporated BBOA material to be further oxidized in the gas-phase. I also think you need to evaluate if not at least part of the reason why the model underestimates the OA is because it underestimates or maybe not even considers biogenic SOA formation.

We thank the reviewer for appreciating our model results. In the revised manuscript we improved the description of methods and added sensitivity tests to give a more comprehensive picture of this model application. The comparison of the results with those from Koo et al. (2014) clearly shows the improvement in the model performance (Fig. S4).

Minor specific comments:

L47, Page 1: Here you use the term "transportation precursors". I think you mean precursors from the road transportation sector. I think you should change the formulation a bit to make this clearer.

Done.

L78-79 You write: "*Moreover, numerous ambient studies of open burning plumes from aircraft do not show a net increase in OA, despite observing oxidation (Cubison et al., 2011; Jolleys et al., 2012).*"
I suggest that you reformulate this sentence and instead write something like: *Moreover, numerous ambient studies with aircraft of open biomass burning plumes do not show a net increase in OA, despite observed oxidation (Cubison et al., 2011; Jolleys et al., 2012).*
When I first read this sentence I thought the open burning plumes came from the aircraft but then I realized that the aircrafts where only used for the measurements of the open biomass burning plumes.

Done.

L98-103: The sentence: *"Ciarelli et al. (2016a) showed that allowing for evaporation of primary organic particles as available in European emission inventories degraded OA performance (further under-predicted OA but with the POA to SOA ratio in a better agreement) whereas model performance improved when volatility distributions that implicitly account for missing semi-volatile material (increasing POA emissions by a factor of 3) were deployed."* is hard to understand. I suggest that you split it into two or three sentences. What do you mean with *"degraded OA performance"*? Do you mean: degraded the model performance concerning the modeled total OA mass?

Yes. We modified the sentence to make it clearer:
*Ciarelli et al. (2016a) showed that allowing for evaporation of primary organic particles as available in the European emission inventories degraded the model performance for the total OA mass (further under-predicted OA but the POA to SOA ratio in a better agreement with measurements). In the same study, on the other hand, model performance improved when volatility distribution that implicitly accounts for missing semivolatile material (increasing POA emissions by a factor of 3) was deployed.*

On L112-115 you write: *"This indirect accounting of missing organic material could be used in the absence of more detailed gridded emission inventories, keeping in mind that the amount of higher volatility compounds was specifically derived from studies conducted with diesel engines (Robinson et al., 2007)."*
In fact I think the Robinson et al., (2007) study was only performed on one single diesel engine (a single-cylinder Yanmar diesel generator), which I expect do not represent modern diesel car engines very well.
I suggest that you instead of "*diesel engines*" at least write: a *singe diesel engine*.

The sentence was modified as:
*This implies that, in these applications, the new emitted organic mass (POA + SVOCs + IVOCs) is 7.5 times higher than in original emissions (i.e., OM = (3\*POA) + (1.5\*(3\*POA))) which could be used as an indirect method to account for missing organic material in the absence of more detailed gridded emission inventories.*
.

On L284-287 you write: *"On the other hand, the remote station of Mace Head showed a positive bias for SOA (MFB = 30%), even though model and measurement concentrations were very similar (0.54 and 0.35 µg m-3, respectively), which could be*
*attributed to an overestimated contribution from the boundaries."*
What do you mean by *"overestimated contribution from the boundaries"*? Is it the influence from the model boundary conditions?

Yes, in this case from the western boundary of the model domain.
On L337-340 you write: *"The model results indicate that non-residential combustion and transportation precursors contribute about 30-40% to SOA formation (with increasing contribution at urban and near-industrialized sites) whereas residential combustion (mainly related to wood burning) contribute to a larger extent, i.e., around 60-70%."*
I suggest that you change to:
*The model results indicate that non-residential combustion and transportation precursors contribute to about 30-40 % of the SOA formation (with increasing contribution at urban and near-industrialized sites) whereas residential combustion (mainly related to wood burning) contribute to a larger extent, i.e., around 60-70%.*

We agree and changed the sentence as suggested.

On line L349-351 you write: *"In the southern part of the domain, the higher temperature will favour more organic material in the semi-volatile range to reside in the gas-phase, rendering it available for oxidation."*
I would also expect that the higher UV-light intensity in the south caused more SOA formation because of higher OH concentrations.

We agree with the referee. In the southern part of the domain more OH should be available to react with secondary organic aerosol precursors. We revised the sentence at line L366-368 as below:

*In the southern part of the domain, where more OH is available, the higher temperature will favour more organic material in the semi-volatile range to reside in the gas-phase, rendering it available for oxidation.*

On line L351-351 you write: "*On the other hand, no south-to-north gradient was predicted for the higher volatility class of precursors.*"
Do you mean?
O*n the other hand, no south-to-north gradient was predicted for the SOA formed from the higher volatility class of precursors.*

Yes. We thank the referee for this comment and we corrected the sentence as suggested.

On L 291-294 you write: "*Mostly traffic-related HOA was underestimated at the urban site Barcelona (Table S2, Fig. 6), with the model not able to reproduce the diurnal variation of HOA at this urban site likely due to poorly reproduced meteorological conditions or too much dilution during day time in the model (Fig. S2).*"
Can it not also be because of too weak diurnal variations in the HOA emissions from traffic in the model?
Reflection: But in the case of Helsinki it seem as if the model instead gives substantially more HOA during the morning (6 UTC, 8 am local time, and 15 UTC, 5 pm local time), which is what you would expect if the HOA mainly came from the local traffic. But surprisingly to me the observations do not indicate any increased local HOA contribution during the morning and afternoon rush hours in Helsinki. Could it be related to the vehicle fleet in Helsinki (i.e. is the road traffic very much dominated by gasoline cars which do not emit much primary HOA but precursors for SOA formation) ?

HOA in Barcelona as determined by PMF analysis displays an atypical diurnal variation with a late peak in the morning and no clear increase in the night. The reason for this behavior is still unknown and is not captured by the model. The site of Barcelona is located in a complex area, i.e. on the coast and close to mountains, which is difficult to model with such a coarse model resolution (0.25x0.25 deg).
On the other hand, it is possible that emissions for Helsinki are not realistic in the model (Karvosenoja et al., 2008). Fountoukis et al. (2014) also reported similar over-prediction at the site of Helsinki for the primary organic fraction during the February- March 2009 period.

**REFERENCES**

Ciarelli, G., Aksoyoglu, S., Crippa, M., Jimenez, J.-L., Nemitz, E., Sellegri, K., Äijälä, M., Carbone, S., Mohr, C., O';Dowd, C., Poulain, L., Baltensperger, U. and Prévôt, A. S. H.: Evaluation of European air quality modelled by CAMx including the volatility basis set scheme, Atmos. Chem. Phys., 16 (16), 10313–10332, doi:10.5194/acp-16-10313-2016, 2016a.

Ciarelli, G., El Haddad, I., Bruns, E., Aksoyoglu, S., Möhler, O., Baltensperger, U. and Prévôt, A. S. H.: Constraining a hybrid volatility basis set model for aging of wood burning emissions using smog chamber experiments, Geosci. Model Dev. , accepted, 2016b.

Crippa, M., DeCarlo, P. F., Slowik, J. G., Mohr, C., Heringa, M. F., Chirico, R., Poulain, L., Freutel, F., Sciare, J., Cozic, J., Di Marco, C. F., Elsasser, M., Nicolas, J. B., Marchand, N., Abidi, E., Wiedensohler, A., Drewnick, F., Schneider, J., Borrmann, S., Nemitz, E., Zimmermann, R., Jaffrezo, J.-L., Prévôt, A. S. H., and Baltensperger, U.: Wintertime aerosol chemical composition and source apportionment of the organic fraction in the metropolitan area of Paris, Atmos. Chem. Phys., 13, 961-981, doi:10.5194/acp-13-961-2013, 2013.

Donahue, N. M., Chuang, W., Epstein, S. A., Kroll, J. H., Worsnop, D. R., Robinson, A. L., Adams, P. J. and Pandis, S. N.: Why do organic aerosols exist? Understanding aerosol lifetimes using the two-dimensional volatility basis set, Environ. Chem., 10(3), 151, doi:10.1071/EN13022, 2013.

Fountoukis, C., Megaritis, A. G., Skyllakou, K., Charalampidis, P. E., Pilinis, C., Denier van der Gon, H. A. C., Crippa, M., Canonaco, F., Mohr, C., Prévôt, A. S. H., Allan, J. D., Poulain, L., Petäjä, T., Tiitta, P., Carbone, S., Kiendler-Scharr, A., Nemitz, E., O'Dowd, C., Swietlicki, E. and Pandis, S. N.: Organic aerosol concentration and composition over Europe: insights from compari-son of regional model predictions with aerosol mass spectrometer factor analysis, Atmos. Chem. Phys., 14(17), 9061-9076, doi:10.5194/acp-14-9061-2014, 2014.

Gantt, B., Johnson, M. S., Crippa, M., Prévôt, A. S. H., and Meskhidze, N.: Implementing marine organic aerosols into the GEOS-Chem model, Geosci. Model Dev., 8, 619-629, doi:10.5194/gmd-8-619-2015, 2015.

Hodzic, A., Kasibhatla, P. S., Jo, D. S., Cappa, C. D., Jimenez, J. L., Madronich, S., and Park, R. J.: Rethinking the global secondary organic aerosol (SOA) budget: stronger production, faster removal, shorter lifetime, Atmos. Chem. Phys., 16, 7917-7941, doi:10.5194/acp-16-7917-2016, 2016.

Jo, D. S., Park, R. J., Kim, M. J., Spraklen, D. V., Effects of chemical aging on lobal secondary organic aerosol using the volatility basis set approach, Atmos., Environ., 81, 230-244, 2008

Karvosenoja, N., Tainio, M., Kupiainen, K., Tuomisto, J. T., Kukkonen, J. & Johansson, M. Evaluation of the emissions and uncertainties of PM2.5 originated from vehicular traffic and domestic wood combustion in Finland. Boreal Env. Res. 13: 465-474, 2008.

Koo, B., Knipping, E. and Yarwood, G.: 1.5-Dimensional volatility basis set approach for modeling organic aerosol in CAMx and CMAQ, Atmos. Environ., 95, 158-164, doi:10.1016/j.atmosenv.2014.06.031, 2014.

---

## Author Comment (AC2) · 4 Apr 2017

**Responses to the comments of anonymous referee #2**

Thank you for your comments that helped to improve our manuscript. Please find below your comments in blue, our responses in black and modifications in the revised manuscript in *italic.*

Ciarelli et al. follow up two other recent publications by augmenting the CAMx VBS implementation with their new parameterization for emission and aging of BBOA emissions. The study itself is a useful application and soundly conceived. The authors find better model-measurement agreement than their previous implementation, but I am troubled by some aspects of their methods and analysis, as described below. Their inclusion of the factor of 3 multiplier to account for missing SVOCs was an approach originally recommended for Mexico City but has not been used for Europe by previous EUCAARI model studies (e.g. Fountoukis et al., 2014). I am open to the authors' interpretation/justification for this choice (especially if I've misinterpreted the situation), but on its face this is a rather critical assumption that could put major aspects of the paper's conclusions in jeopardy. Moreover, the application of modeled PM2.5 mass to PM1.0 measurements raises questions about how much of the model agreement is spurious. Considering both of these potential biases together, it is concerning that the model predictions for SOA and POA are still lower in many cases than the VBS predictions published by Fountoukis et al. (2014) for the same model scenario. I could recommend this paper for publication after these issues are resolved.

Specific comments:

1. Page 4, line 108-113: The ratio of semivolatile to nonvolatile material is, as the authors know, a function of the emission source, fuel, and operating conditions – I think it is overly simplistic and actually unhelpful to state that the ratio is predicted to be "roughly 3." The Shrivastava et al. (2011) and Tsimpidi et al. (2010) studies argued that those SVOCs at Mexico City were missing from the inventories because the emissions were parameterized using ambient observations of OA, which would have already equilibrated to atmospheric conditions. On the other hand, the emission factors used to inform the gridded inventories of Europe and the US are, to my knowledge, derived from laboratory scale tests, where much of those SVOCs are notoriously condensed in the particle phase in undiluted exhaust. My reading of Fountoukis et al. (2014) does not lead me to believe that they enhanced their SVOC emissions by a factor of 3 over POA. Rather, I believe they simply repartitioned the existing POA, and they added an additional 1.5*POA for the IVOCs as the authors state. Ciarelli et al. (2016a) shows that the extra SVOCs are needed to improve the model performance (i.e. VBS_BC did much better than VBS_ROB), but I disagree that there is evidence that SVOCs are underestimated in European inventories by so much. Instead, I would argue the real source of this mass is still unknown and is probably a combination of underestimated SOA yields, aqueous processing, aging of anthropogenic and biogenic SOA and some missing SVOCs as well.

At minimum, a considerable amount of rewriting in the methods, conclusions and abstract is necessary so that the authors communicate explicitly that an unknown fraction of these SVOCs are very likely double-counted and that this parameter needs to be refined and probably lowered in the future as more explicit pathways are added to the model.

It is true that our previous studies indicated a deterioration of the model performance for OA when evaporation of primary organic particles was allowed while using the approach proposed for the Mexico city study (Shrivastava et al. (2011); Tsimpidi et al. (2010)) led to a better performance (VBS_BC in Ciarelli et al., 2016a). We agree with the referee that other factors such as underestimation of SOA yields, aqueous processing and aging of anthropogenic and biogenic SOA might also play a crucial role in addition to missing SVOCs and we also agree that the factor of 3 used in this study for the inclusion of semi-volatile organic compounds might have high uncertainties.

This choice however, was based on the recent European study by Denier van der Gon et al. (2015) rather than the Mexico City studies. The OA emissions in Europe have often been claimed to be underpredicted in current inventories (Bergstrom et al., 2012; Fountoukis et al., 2014) but only recent studies are starting to better elucidate the range of uncertainties related to them, in particular because of the semi-volatile nature of such material. In the work of Denier van der Gon et al. (2015) a revised residential wood combustion (named TNO-newRWC) emission inventory was compiled for Europe using a bottom-up approach. The authors underlined the importance of various sampling methods and measurement protocols or techniques influencing particle emission factors using data from the survey of Nussbaumer et al. (2008a,b). The most important sampling methods used by the countries participating in the survey, were filter measurements of only solid particles (SP) and dilution tunnel (DT) measurements of solid particles and condensable organics (or semivolatile organics). For conventional woodstoves, the authors found a difference in PM emission factors by a factor of up to 5 between the two techniques. The revised emission inventory (TNO-newRWC) was later compiled using the average DT emission factor from different type of appliance (Table 2 in Denier van der Gon et al., 2015) and compared with previously used emission inventory EUCAARI. The authors concluded that the revised emissions were higher than those in the EUCAARI inventory by a factor of 2-3 which is similar to the correction factor used in our study and in Shrivastava et al. (2011) and Tsimpidi et al. (2010). However, it should be noted that a substantial inter-country variation was reported within the gridded emission inventories which might lead to over or underestimation of emissions depending on the country (for example the ratio between the TNO-newRWC and EUCAARI emission inventory was around 1-3 in France and up to a factor of 5-10 in Sweden and Finland).

Denier van der Gon et al. (2015) also used the revised emission inventory in two commonly used chemical transport models carrying the VBS scheme to perform the organic chemistry: PMCAMx and EMEP model. They found that the revised emission inventory substantially improved the agreement between measured and predicted organic aerosol for the same period presented in this study (Feb-Mar 2009) with results in line with the VBS_BC scenario performed in Ciarelli et al. (2016a) and VBS_BC_NEW presented here. Therefore, we think that the correction factor proposed in this study (factor of 3) can be used until detailed emission inventories including semivolatile compounds are available for the modeling community. Moreover, other explicit SOA formation pathways must be included as more experimental data will be available.

We inserted the following statements in the abstract, conclusions and method part as suggested by the referee.

In the abstract as below:

*Although the new parameterization leads to a better agreement between model results and observations, it still under-predicts the SOA fraction suggesting that uncertainties in the new scheme and other sources and/or formation mechanisms remain to be elucidated. Moreover, a more detailed characterization of the semivolatile components of the emissions is needed.*

In the method at line 171 of the revised manuscript:

*In order to include gas-phase organics in the semivolatile range in the absence of more detailed inventory data, we used the approach of increasing the standard emissions by a factor of 3 proposed by previous studies (Shrivastava et al., 2011; Tsimpidi et al., 2010) which is also in line with the recent European study on the revision of the residential wood combustion emissions (Denier van der Gon et al., 2015). This approach of including the semivolatile compounds can be used until detailed emission inventories with more realistic inter-country distribution of the emissions become available (e.g. Denier van der Gon et al., 2015).*

In the conclusions, line 417 of the revised manuscript:

*On the other hand, the modelled BBPOA was higher than the measurements at several stations indicating the need for further studies on residential heating emissions, their volatility distribution and oxidation pathway of the semivolatile organic gases. In addition, more detailed emission inventories are needed to characterize the semivolatile components better, as proposed by Denier van der Gon et al. (2015).*

2. I agree with the first reviewer that there needs to be significant more description of the VBS framework used here. The diagrams in Ciarelli et al. (2016b) are helpful and there should be a table or diagram in this manuscript that summarize that information for the entire VBS picture including emissions and aging.

We agree and revised the part about the VBS scheme as suggested also by Referee #1 in Section 2.2 as below. We added Table 1 to summarize the description of the VBS spaces.

*2.2 Organic aerosol scheme*

[revised manuscript text omitted]

3. What is being done about wildfires in the model? Were there any during the EUCAARI scenario? Are they represented well in the emissions inputs? If so, how do they effect the source apportionment analysis that is presented?

Emissions from wildfires were not considered for this application since they were not delivered in the EURODELTA3 exercise for the year 2009 (Bessagnet et al., 2016). We analyzed the fire emission data (non-domestic fires) as available in IS4FIRES data (Sofiev et al., 2009) obtained by re-analysis of fire radiative power data from the MODIS instrument. Figure 1 below shows the cumulative emissions from wildfires in kg/s in March 2009. Significant fires occurred mainly in the north of Portugal. We think that the effect on the simulated OA concentrations might be quite limited since all the investigated stations are located quite far from that area.

[Figure]

*Figure 1. Cumulative emissions from wildfires in kg/s during the month of March 2009 as in IS4FIRES (Sofiev et al., 2009).*

4. On page 5, lines 150-151, the authors point out that CAMx is predicting PM2.5. But the evaluation is against AMS observations which I presume are primarily PM1.0. Doesn't this fact make the frequent underprediction in SOA even more troubling? Is anything more specific known about the diameter of PM2.5 particles to allow the authors to estimate the fraction that would be PM1.0 and thus more applicable to the measurements?

This issue was discussed in Aksoyoglu et al. (2011) where $PM_1$ and $PM_{2.5}$ measurements in Payerne during both winter (January 2007) and summer (June 2006) periods were compared. The authors concluded that the differences between the two fractions were usually rather small (Figure 14 in Aksoyoglu et al., 2011). These results are also supported by a recent study where comparisons between the organic matter concentrations in $PM_1$ and $PM_{2.5}$ fractions in winter were found to be in the same range (Bozzetti et al., 2016).

5. Given that points 1 and 4 would lead one to expect substantial overprediction by the model, please also explain why the current predictions are lower than those in Fountoukis et al. (2014) at many sites.

We don't expect substantial overprediction due to particle size as we explained above at point 4. Even though both model simulations (this study and Fountoukis et al., 2014) mostly cover the same domain and time period, some differences are expected due to model resolution, different input data used in simulations as well as the differences in chemical mechanisms. Fountoukis et al. (2014) account for marine OA emissions which are not included in our simulations. In Fountoukis et al. (2014), the emissions were calculated based on the scheme of O'Dowd et al. (2008) and the organic aerosol fractions allocated in both fine and coarse mode. Gantt et al., (2015) showed that primary marine organic aerosol have a weaker coastal-to-inland concentrations gradient than sea-salt aerosol with some inland European cities having more than 10% of the submicron organic aerosol mass as a marine source. Differences in fire emissions, even if present, are likely to be less important for this comparison since there were few fires activity data during the considered periods (as addressed in comment 3).

Another difference is the way the boundary conditions for OA are taken into account. Fountoukis et al. (2014) used fixed boundary conditions based on measured average background concentrations in sites close to the boundaries of the domain whereas we derived OA boundary fields from MACC reanalysis data (Inness et al., 2013; Benedetti et al., 2009). In our study, OA fields at the domain boundaries are distributed as half –half between POA and SOA, as prescribed by the EURODELTA3 exercise, whereas in Fountoukis et al. (2014) OA at the boundaries are assumed to be all oxidized (SOA).

In addition to the input data, different gas-phase mechanisms used in both studies (CB05 in this study,

SAPRC99 in Fountakis et al., 2014) might lead to different results. One has also to keep in mind the different grid resolution ($0.25^0$ x $0.25^0$ in this study, 36kmx36km in Fountoukis et al., 2014) while comparing the two studies.

6. Page 9, lines 269-272: This discussion of Fig. 5 is very light. If there is not more to discuss, I recommend removing the figure and just stating the improvement in MB and r

We prefer to keep Fig. 5 since the figure allows the readers to see the differences for POA and SOA. We combined the two paragraphs where the discussion involves Fig. 5 as follows in line 284-294 of the revised manuscript:

*Comparison of results from this study (VBS_BC_NEW) with the earlier one (VBS_BC, Ciarelli et al., 2016a) suggests that the new VBS scheme predicts higher SOA concentrations by about a factor of 3 (Fig. 5) and improves the model performance when comparing assessed OOA from measurements with modelled SOA (Table 4). POA concentrations, on the other hand, are clustered below 1 µg m$^{-3}$ except in Barcelona (Fig. 5), showing an $R^2$=0.36 (Table 3). Although predicted POA concentrations at Barcelona were lower than the measurements, MFB=-47% and MFE=69% were still in the range for acceptable performance criteria (MFE $\leq$+75% and $-60 < MFB < + 60$ %, Boylan and Russell, 2006). On the other hand, the model over-predicted the POA concentrations at Hyytiälä (MFB=131% and MFE=131%), Helsinki (MFB=95% and MFE=100%) and Cabauw (MFB=76% and MFE=86%) mainly due to the overestimated BBPOA fraction as seen in Fig. 6.*

7. How does the BBOA doubling sensitivity case fit in the context of the VBS_BC_NEW case which is multiplied by 3 and then by 1.5 again? What fraction of that total added vapor mass makes it into the particle phase? This is related to point 8

The BBOA doubling sensitivity case performed in Ciarelli et al. (2016a) with the original VBS scheme (Koo et al., 2014) was reported in order to show that although the model performance for total OA concentration improved significantly, that was not the case for the OA components, with POA being over-predicted at almost all the sites and no significant effect was observed on the modelled SOA concentrations (Figure 10). The rest of the comment is addressed in comment Nr. 8 (below).

8. The description and discussion of BBOA aging should be expanded. Please summarize the aging process as described in Ciarelli et al. (2016b). How is this similar/different to the aging of the traditional biogenic SOA? I assume the authors are not using the Koo et al. (2014) approach where the BBOA ages once and then stops ? What is the fractional contribution of the various volatility bins to the total in time and space? Do they actually need 4 VBS bins to represent the aging, or would just using one bin and an IVOC precursor also work reasonably well? Why did they not use the O:C obtained from these AMS data to constrain the aging of the BBOA or the SOA?

A total number of 3 sets were used to describe the evolution of organic material. The first set was used to distribute the primary emissions (set1). Two other sets were used to model the formation and evolution of SOA. Oxidation products of SVOC material arising from primary emissions were allocated to set2, whereas oxidation products from NTVOCs (non-traditional VOCs) were allocated to set3. The specific molecular structures for each of the sets and bins were retrieved using the group contribution approach and the Van Krevelen relation (Donahue et al., 2011; Heald et al., 2010). The oxidation of semi-volatile material would tend to increase the compounds' oxygen number and decrease their volatility and carbon number, due to functionalization and fragmentation. We assume that the oxidation of the primary semi-volatile compounds with C11-C14 decreases their volatility by one order of magnitude and yields C9-C10 surrogates, placed in set2, based on the work of Donahue et al. (2011, 2012). Based on these assumptions and using the group contribution approach, the oxygen numbers for set2 is predicted to vary between 2.26 and 4.56. Thus, the model implicitly accounts for the addition of 1.1 to 1.5 oxygen atoms and the loss of 2.75 to 4.25 carbon atoms, with one oxidation step. Set3 was constrained based on the PTR-MS data. The measurements suggested an average NTVOC carbon and oxygen number of about 7 and 1, respectively. Based on reported molecular speciation data (e.g. Kleindienst et al., 2007), we expect the products of C7 compounds to have a C5-C6 carbon backbone. These products were placed in set3 following a kernel function based on the distribution of

naphthalene oxidation products. At least two oxygen atoms were added to the NTVOC mixture upon their oxidation. The overall O:C ratio in the whole space roughly spans the range from 0.1 to 1.0. Multigeneration chemistry (aging) is also accounted for by the model. Unlike the 2D-VBS, the 1.5D-VBS does not use different kernel functions, to discretize the distribution of the oxidation products according to their log(C*) and O:C ratios, when functionalization and fragmentation occur. Instead, to reduce the computational burden of the simulations, the model assumes that the oxidation of a given surrogate yields one other surrogate with lower volatility, higher oxygen number and lower carbon number. These properties should be considered as a weighted average of those relative to the complex mixture of compounds arising from functionalization and fragmentation processes. Accordingly, the 1.5D-VBS approach represents the functionalization and fragmentation processes effectively while reducing the parameter space and the computational burden. Gas-phase products in the semi-volatile range in set2 and set3, once formed, can further react with a rate constant of $4 \times 10^{-11}$ cm3 molecule-1 s-1 as proposed by previous studies (Donahue et al., 2013; Grieshop et al., 2009; Robinson et al., 2007), further lowering the volatility of the products by one order of magnitude. This implies that for every additional oxidation step, the organic material receives around 0.5 oxygen atoms.

However, we don't have a separate set to allocate oxidation products from biogenic precursors, and they follow the same oxidation pathways of biomass burning like aerosol as in the previous case (Ciarelli et al., 2016a), including aging. We are currently working on an updated version of CAMx that includes the separation of biogenic sources.

Minor Issues/Typos

1. Page 2, line 53: What do the authors mean by "higher volatility?" Are these IVOCs or VOCs? And do they mean that the products of these and the semivolatile precursors contributed 15 to 38%?

Higher volatility refers to IVOCs and VOCs. Only the products of IVOCs and VOCs contributed to 15 to 38%. We rephrase for clarification as below:

*On the other hand, the oxidation products of higher volatility precursors (the sum of IVOCs and VOCs) contribute from 15 to 38% with no specific gradient among the stations.*

2. Page 3, line 62: Consider replacing "qualitatively" with "nominally." They are very similar for sure but while qualitatively to me suggests one knows a lot about the relative importance of each source (just not the actual numbers), nominal suggests you just know that the sources are there and you can name them. The latter to me is more representative of our knowledge of sources for SOA.

Done.

3. Page 3, lines 65-71: Please also mention aqueous-phase formation and the importance of solubility in water somewhere here to make the picture more complete.

We added the following sentence at line 70-75 of the revisited manuscript to mention the importance of aqueous-phase formation as below:

*The physical and chemical processes leading to the formation of SOA are numerous, e.g. oxidation and condensation, oligomerization or aqueous-phase formation, and they are very uncertain and currently under debate (Hallquist et al., 2009; Tsigaridis et al., 2014; Fuzzi et al., 2015; Woody et al., 2016). Moreover, the solubility of organic compounds in water is also a crucial parameter affecting the life time of organic particles and gases in the atmosphere (Hodzic et al., 2016).*

4. Page 3, line 82: Consider removing the word "common." And refer to SOA explicitly here. For example: "Most CTMs today account for SOA formation from biogenic and anthropogenic...A few models also include SOA formation from intermediate volatility„".

Done.

5. I don't think you need a hyphen in "semi-volatile" anywhere in the text, but this is your preference.

Done.

6. Page 4, line 114-115: The higher volatility emission parameters were also constrained using monitoring network measurements in the previous modeling studies. Several studies have played with 1.5 factor for instance and it has remained as the parameter of choice despite uncertainties.
We thank the referee for this comment. We reformulated and shortened the sentence from line 116-119 of the revised manuscript as below:

*This implies that, in these applications, the new emitted organic mass (POA + SVOCs + IVOCs) is 7.5 times higher than in original emissions (i.e., OM = (3\*POA) + (1.5\*(3\*POA))) which could be used as an indirect method to account for missing organic material in the absence of more detailed gridded emission inventories.*

7. Page 7, lines 193-199: I was confused by this group of sentences. Consider rewriting for clarity. Maybe something like, "We assumed OA emissions from SNAP2 (emissions from non-industrial combustion plants in the Selected Nomenclature for Air Pollution) and SNAP10 (emissions from agriculture, about 6% of POA in SNAP2), to be representative of biomass burning emissions and thus comparable to the BBOA PMF factor. OA from all other SNAP categories were compared against HOA-like PMF factors. Unfortunately, gridded emissions for SNAP2 include other emission sources (i.e., coal burning which might be important in eastern European countries like Poland). We could not resolve our emission inventory with sufficient detail to separate the contribution of coal for these European cites (Crippa et al., 2014)."

We agree and modified the sentence as suggested by the referee (line 205 in the revised manuscript).

8. Page 8, line 219: Please do not call it deposition "capacity" as this suggests something about the ability of the sea to hold pollution. Please reword. "Efficiency" might make more sense. Or just say "reduced deposition". Also change on page 9, line 267

Done.

9. Page 8, line 236: Please provide some statistic for this statement.

We added the statistics: ($R^2$=0.72).

10. Fig. 3: Consider adding error bars to this plot showing variability to make this figure more useful.

Done.

[Figure]

*Figure 3. Observed (black) and modelled (VBS_BC_NEW) (red) average OA mass at AMS sites for the period between 25 February and 26 March 2009.*

**11. Page 9, lines 258-262: This sentence needs to be split into two sentences and reworded for clarity.**

We reworded the sentences at line 274-278 of the revised manuscript as below:

*Bergström et al. (2012) reported that emissions of organic carbon (OC) from the residential heating sector in Sweden were lower than those in Norway by a factor of 14 in spite of its higher wood usage by 60%. This indicates an underestimation of emissions from residential heating in the emission inventory.*

**12. Page 10, line 288-290: Do you have evidence from other PM species or pollutants to back up this claim?**

We added a comparison for the modelled PM$_{25}$ components for Puy de Dome and Montseny. At both sites all the components were over-predicted (Fig. S3). We added the following sentence at line 304 of the revised manuscript.

*as confirmed by the over-prediction of other PM species at these sites (Fig. S3).*

[Figure]

*Figure S3. Comparison of modelled non-refractory PM$_{25}$ components at Puy de Dome and Montseny with the AMS measurements in February-March 2009.*

**13. Page 10, line 291-305: This sentence should be revised for clarity. The authors have blamed the meteorology and the host model configuration itself but why not the emissions? The activity data for the emissions could be wrong, or the emission factors could be wrong, no? Ok, CAMx has issues like any other CTM, but what makes the authors so sure that most of the problem is not in the emissions data?**

We agree with the referee. Emissions might also represent a large source of uncertainties, recently, even more than previously thought. We added more emphasis on this point at line 311-315 of the revised manuscript as below:

*In addition, the gridded emission inventories still represent a large source of uncertainties for CTM applications. The majority of the NOx (NO+NO$_2$) emissions in Europe arises from the transportation sector (SNAP7), which might have much larger uncertainties than previously thought (Vaughan et al., 2016).*

**14. Page 10, line 296: course should be spelled coarse**

We corrected the typo.

**15. Page 10, line 308-315: The authors can also add here the potential double counting of SVOC emissions and the application of PM2.5 prediction to a (nominal) PM1.0 measurement**

We modified the sentence from line 330 of the revised manuscript in order to include other reasons for the over prediction of the BBOA fraction as below:

*4) The simplistic way of accounting for the semi-volatile part of primary emissions might lead, in some areas, to the double counting of such compounds.*
*5) Uncertainties in the retrieved BBOA fraction from PMF analysis.*

16. Page 11, line 316-318: How many of the peaks were captured well? What statistic determines how well they were captured? Unless this statement can be quantified, please remove it.

We removed the sentence.

17. Page 11, line 322: Please consider changing "likely" to "possibly."

Done.

18. Figure 10. Please consider using median values in these plots rather than averages. 1) It will more effectively reduce the influence of extreme pollution days. 2) It will be more consistent with your use of percentiles. Consider also adding percentiles for the model run data.

Done.

[Figure]

*Figure 10. POA (left) and SOA (right) median concentrations at 8 AMS sites for February-March 2009 in the VBS_BC, VBS_BC_2xBBOA and VBS_BC_NEW cases. Dotted lines indicate the 10th and 90th quartile range (also reported in red for the VBS_BC_NEW case). Data for the Puy de Dôme and Montseny sites at higher layers are not available for the VBS_BC_2xBBOA scenario.*

19. Figure 11: This data would be better represented as a bar plot since the x-axis is not really a continuum, even though you are trying to approximate one by ordering them south-north.

We agreed and replaced the Figure 11 with a bar plot.

20. Tables: please add one more significant figure to all data. I can't figure out why the mean biases are different than the differences in the mean model and mean obs. Is it a rounding issue?

The observed differences are indeed due to the rounding. However, we prefer to keep one significant digit in the tables.

21. Page 13, line 380-388: Please quantify "reasonably good." Compared to what?

We modify the sentence at line 413 as below:

*Predicted HOA concentrations were in the range of those retrieved from the PMF analysis..*

22. Figure 11: Is BBOA actually just primary BBOA? Please make this clear in this figure and throughout the text as it gets confusing.

We thank the reviewer for this comment that was also addressed by referee #1. Yes, BBOA refers only to primary BBOA. We changed the legend in Figure 11 as presented in comment 19. We changed BBOA to BBPOA in the whole manuscript.